# Characterizing Graph Datasets for Node Classification: Homophily–Heterophily Dichotomy and Beyond

**Oleg Platonov**[*]
HSE University
Yandex Research
olegplatonov@yandex-team.ru

**Denis Kuznedelev**
Yandex Research
Skoltech
dkuznedelev@yandex-team.ru

**Artem Babenko**
Yandex Research
artem.babenko@phystech.edu

**Liudmila Prokhorenkova**[*][†]
Yandex Research
ostroumova-la@yandex-team.ru

## Abstract

*Homophily* is a graph property describing the tendency of edges to connect similar nodes; the opposite is called *heterophily*. It is often believed that heterophilous graphs are challenging for standard message-passing graph neural networks (GNNs), and much effort has been put into developing efficient methods for this setting. However, there is no universally agreed-upon measure of homophily in the literature. In this work, we show that commonly used homophily measures have critical drawbacks preventing the comparison of homophily levels across different datasets. For this, we formalize desirable properties for a proper homophily measure and verify which measures satisfy which properties. In particular, we show that a measure that we call *adjusted homophily* satisfies more desirable properties than other popular homophily measures while being rarely used in graph machine learning literature. Then, we go beyond the homophily–heterophily dichotomy and propose a new characteristic that allows one to further distinguish different sorts of heterophily. The proposed *label informativeness* (LI) characterizes how much information a neighbor's label provides about a node's label. We prove that this measure satisfies important desirable properties. We also observe empirically that LI better agrees with GNN performance compared to homophily measures, which confirms that it is a useful characteristic of the graph structure.

## 1 Introduction

Graphs are a natural way to represent data from various domains such as social networks, citation networks, molecules, transportation networks, text, code, and others. Machine learning on graph-structured data has experienced significant growth in recent years, with Graph Neural Networks (GNNs) showing particularly strong results. Many variants of GNNs have been proposed [16, 10, 43, 45], most of them can be unified by a general Message Passing Neural Networks (MPNNs) framework [7]. MPNNs combine node features (attributes) with graph topology to learn complex dependencies between the nodes. For this, MPNNs iteratively update the representation of each node by aggregating information from the previous-layer representations of the node itself and its neighbors.

---

[*]Equal contribution
[†]Corresponding author

37th Conference on Neural Information Processing Systems (NeurIPS 2023).

In many real-world networks, edges tend to connect similar nodes: users in social networks tend to connect to users with similar interests, and papers in citation networks mostly cite works from the same research area. This property is usually called *homophily*. The opposite of homophily is *heterophily*: for instance, in social networks, fraudsters rarely connect to other fraudsters, while in dating networks, edges often connect the opposite genders. Early works on GNNs mainly focus on homophilous graphs. However, it was later discovered that classic GNNs typically do not perform well on heterophilous graphs, and new GNN architectures have been developed for this setting [1, 4, 32, 48, 49].

To measure the level of homophily, several *homophily measures* are used in the literature [1, 20, 32, 48], but these measures may significantly disagree with each other. In this work, we start by addressing the problem of how to properly measure the homophily level of a graph. For this, we formalize some desirable properties of a reasonable homophily measure and check which measures satisfy which properties. One essential property is called *constant baseline* and, informally speaking, it requires a measure to be not biased towards particular numbers of classes or their size balance. Our analysis reveals that commonly used homophily measures do not satisfy this property and thus cannot be compared across different datasets. In contrast, a measure that we call *adjusted homophily* (a.k.a. *assortativity coefficient*) satisfies most of the desirable properties while being rarely used in graph ML literature. Based on our theoretical analysis, we advise using adjusted homophily as a better alternative to the commonly used measures.

Then, we note that heterophilous datasets may have various connectivity patterns, and some of them are easier for GNNs than others. Motivated by that, we propose a new graph property called *label informativeness* (LI) that allows one to further distinguish different sorts of heterophily. LI characterizes how much information the neighbor's label provides about the node's label. We analyze this measure via the same theoretical framework and show that it satisfies the constant baseline property and thus is comparable across datasets. We also observe empirically that LI better agrees with GNN performance than homophily measures. Thus, while being very simple to compute, LI intuitively illustrates why GNNs can sometimes perform well on heterophilous datasets — a phenomenon recently observed in the literature. While LI is not a measure of homophily, it naturally complements adjusted homophily by distinguishing different heterophily patterns.

In summary, we propose a theoretical framework that allows for an informed choice of suitable characteristics describing graph connectivity patterns in node classification tasks. Based on this framework, we suggest using adjusted homophily to measure whether similar nodes tend to be connected. To further characterize the datasets and distinguish different sorts of heterophily, we propose a new measure called label informativeness.

## 2 Homophily measures

Assume that we are given a graph $G = (V, E)$ with nodes $V$, $|V| = n$, and edges $E$. Throughout the paper, we assume that the graph is simple (without self-loops and multiple edges) and undirected.[3] Each node $v \in V$ has a feature vector $\mathbf{x}_v \in \mathbb{R}^m$ and a class label $y_v \in \{1, \ldots, C\}$. Let $n_k$ denote the size of $k$-th class, i.e., $n_k = |\{v : y_v = k\}|$. By $N(v)$ we denote the neighbors of $v$ in $G$ and by $d(v) = |N(v)|$ the degree of $v$. Also, let $D_k := \sum_{v : y_v = k} d(v)$. Let $p(\cdot)$ denote the empirical distribution of class labels, i.e., $p(k) = \frac{n_k}{n}$. Then, we also define degree-weighted distribution as $\bar{p}(k) = \frac{\sum_{v : y_v = k} d(v)}{2|E|} = \frac{D_k}{2|E|}$.

### 2.1 Popular homophily measures

Many GNN models implicitly make a so-called *homophily* assumption: that similar nodes are connected. Similarity can be considered in terms of node features or node labels. Usually, *label homophily* is analyzed, and we also focus on this aspect, leaving *feature homophily* for further studies. There are several commonly used homophily measures in the literature. *Edge homophily* [1, 48] is the fraction of edges that connect nodes of the same class:

$$h_{edge} = \frac{|\{\{u, v\} \in E : y_u = y_v\}|}{|E|}. \tag{1}$$

---

[3]We further denote (unordered) edges by $\{u, v\}$ and ordered pairs of nodes by $(u, v)$.

*Node homophily* [32] computes the fraction of neighbors that have the same class for all nodes and then averages these values across the nodes:

$$h_{node} = \frac{1}{n} \sum_{v \in V} \frac{|\{u \in N(v) : y_u = y_v\}|}{d(v)} .$$

These two measures are intuitive but have the downside of being sensitive to the number of classes and their balance, which makes them hard to interpret and incomparable across different datasets [20]. For example, suppose that each node in a graph is connected to one node of each class. Then, both edge homophily and node homophily for this graph will be equal to $\frac{1}{C}$. Thus, these metrics will produce widely different values for graphs with different numbers of classes, despite these graphs being similar in exhibiting no homophily. To fix these issues, Lim et al. [20] propose another homophily measure sometimes referred to as *class homophily* [22]. Class homophily measures excess homophily compared to a null model where edges are independent of the labels. More formally,

$$h_{class} = \frac{1}{C-1} \sum_{k=1}^{C} \left[ \frac{\sum_{v:y_v=k} |\{u \in N(v) : y_u = y_v\}|}{\sum_{v:y_v=k} d(v)} - \frac{n_k}{n} \right]_+,$$

where $[x]_+ = \max\{x, 0\}$. The factor $\frac{1}{C-1}$ scales $h_{class}$ to the interval $[0, 1]$; larger values indicate more homophilous graphs and non-homophilous ones are expected to have close to zero values.

However, there are still some issues with class homophily. First, when correcting the fraction of intra-class edges by its expected value, class homophily does not consider the variation of node degrees. Indeed, if nodes of class $k$ have, on average, larger degrees than $2|E|/n$, then the probability that a random edge goes to that class can be significantly larger than $n_k/n$. Second, only positive deviations from $n_k/n$ contribute to class homophily, while classes with heterophilous connectivity patterns are neglected. Let us illustrate these drawbacks of class homophily with a simple example.

**Example** Let us construct non-homophilous graphs for which class homophily is significantly larger than zero. First, we take a clique of size $r$ with all nodes belonging to the red class; then, for each node in the clique, connect it to $r - 1$ leaves, all of which belong to the blue class (example for $r = 4$ is shown on the right). Note that all blue nodes are strictly heterophilous (i.e., only connect to nodes of the opposite class), while all red nodes are class-agnostic (i.e., have the same number of neighbors of both classes). Such graphs are clearly non-homophilous, and a meaningful homophily measure should not produce a value significantly greater than zero for them. However, class homophily for such graphs is positive and can become as large as $\frac{1}{2}$: $h_{class} = \frac{1}{2} - \frac{1}{r} \to \frac{1}{2}$ as $r \to \infty$. 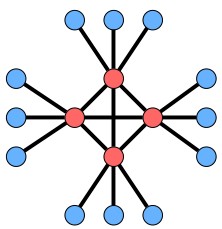

## 2.2 Desirable properties for homophily measures

Above, we discussed some disadvantages of existing homophily measures. In this section, we formalize and extend this discussion: we propose a list of properties desirable for a good homophily measure. Our analysis is motivated by recent studies of clustering and classification performance measures [8, 9], but not all their properties can be transferred to homophily measures. For instance, we do not require *symmetry* — a property that a measure does not change when we swap the compared objects — since homophily compares entities of different nature (a graph and a labeling). For the same reason, the *distance* property (requiring a measure to be linearly transformed to a metric distance) cannot be defined. On the other hand, some of our properties are novel.

**Maximal agreement**    This property requires that perfectly homophilous graphs achieve a constant upper bound of the measure. Formally, we say that a homophily measure $h$ satisfies maximal agreement if for any graph $G$ in which $y_u = y_v$ for all $\{u, v\} \in E$ we have $h(G) = c_{\max}$. For all other graphs $G$, we require $h(G) < c_{\max}$.

**Minimal agreement**    We say that a homophily measure $h$ satisfies minimal agreement if $h(G) = c_{\min}$ for any graph $G$ in which $y_u \neq y_v$ for all $\{u, v\} \in E$. For all other graphs $G$, we require $h(G) > c_{\min}$. In other words, if all edges connect nodes of different classes, we expect to observe a constant minimal value.

**Constant baseline** This property ensures that homophily is not biased towards particular class size distributions. Intuitively, if the graph structure is independent of labels, we would expect a low homophily value. Moreover, if we want a measure to be comparable across datasets, we expect to observe the same low value in all such cases. There are several ways to formalize the concept of independence, and we suggest the one based on the so-called *configuration model*.

**Definition 1.** *Configuration model* is defined as follows: take $n$ nodes, assign each node $v$ degree $d(v)$, and then randomly pair edge endpoints to obtain a graph.[4]

Assuming that we are given $n$ labeled nodes and the graph is constructed according to the configuration model (independently from the labels), we expect to observe a fixed (small) homophily independently of the number of classes and class size balance. We formalize this property as follows and refer to Appendix B.2 for other possible definitions.

**Definition 2.** A homophily measure $h$ has *asymptotic constant baseline* if for $G$ generated according to the configuration model and for any $\varepsilon > 0$ with probability $1 - o(1)$ we have $|h(G) - c_{base}| < \varepsilon$ for some constant $c_{base}$ as $n \to \infty$.

In combination with maximal agreement, asymptotic constant baseline makes the values of a homophily measure comparable across different datasets: the maximal agreement guarantees that perfectly homophilous graphs have the same value, while constant baseline aligns the uninformative cases with neither strong homophily nor strong heterophily.

**Empty class tolerance** Since homophily measures are used to compare different graph datasets, they have to be comparable across datasets with varying numbers of classes. For this, the following property is required.

**Definition 3.** A measure is tolerant to empty classes if it is defined and it does not change when we introduce an additional dummy label that is not present in the data.

For instance, edge homophily and node homophily are empty class tolerant, while class homophily is not. Empty class tolerance is a new property that was not discussed in [8, 9] since classification and clustering evaluation measures are used within one given dataset, see Appendix B.3 for more details.

**Monotonicity** As we discuss in Appendix B.3, it can be non-trivial to define *monotonicity* for homophily measures and there can be different possible options. In this paper, we use the following definition that aligns especially well with edge-wise homophily measures discussed below in Section 2.5.

**Definition 4.** A homophily measure is *monotone* if it is empty class tolerant, increases when we add an edge between two nodes of the same class (except for perfectly homophilous graphs) and decreases when we add an edge between two nodes of different classes (except for perfectly heterophilous graphs).

In contrast to Gösgens et al. [8, 9], our notion of monotonicity requires the property to hold across graphs with different numbers of classes. This is caused by the empty class tolerance property.

### 2.3 Properties of popular homophily measures

Below we briefly discuss the properties of popular homophily measures. In most cases, the proofs are straightforward or follow from Section 2.5 below. Table 1 summarizes the results.

*Edge homophily* satisfies maximal and minimal agreement and is empty class tolerant and monotone. However, it does not satisfy asymptotic constant baseline, which is a critical drawback: one can get misleading results in settings with imbalanced classes.

*Node homophily* satisfies maximal and minimal agreement. It is empty class tolerant, but not monotone: adding an edge between two perfectly homophilous nodes of the same class does not change node homophily. Similarly to edge homophily, node homophily does not satisfy the asymptotic constant baseline and thus is incomparable across different datasets.

---

[4]See Appendix A for additional discussions about the model.

Table 1: Properties of homophily measures: maximal agreement (Max), minimal agreement (Min), asymptotic constant baseline (ACB), empty class tolerance (ECT), monotonicity (Mon). ✗* denotes that the property is not satisfied in general, but holds for large $h_{adj}$ (see Theorem 1).

| Measure | Max | Min | ACB | ECT | Mon |
|---|---|---|---|---|---|
| Edge homophily | ✓ | ✓ | ✗ | ✓ | ✓ |
| Node homophily | ✓ | ✓ | ✗ | ✓ | ✗ |
| Class homophily | ✓ | ✗ | ✗ | ✗ | ✗ |
| Adjusted homophily | ✓ | ✗ | ✓ | ✓ | ✗* |

*Class homophily* satisfies maximal agreement with $h_{class} = 1$, but minimal agreement is not satisfied: not only perfectly heterophilous graphs may have $h_{class} = 0$. Class homophily is not empty class tolerant and thus is not monotone. Additionally, it does not have the asymptotic constant baseline. See Appendix B.1 for the proofs and discussions. Interestingly, removing the $[\cdot]_+$ operation from the definition of class homophily solves the problem with the asymptotic constant baseline, but minimal agreement, empty class tolerance, and monotonicity are still violated.

## 2.4 Adjusted homophily

Let us discuss a much less known homophily measure that by construction satisfies two important properties — maximal agreement and constant baseline. To derive this measure, we start with edge homophily and first enforce the constant baseline property by subtracting the expected value of the measure. Under the configuration model, the probability that a given edge endpoint will be connected to a node with a class $k$ is (up to a negligible term) $\frac{\sum_{v:y_v=k} d(v)}{2|E|}$. Thus, the adjusted value becomes $h_{edge} - \sum_{k=1}^{C} \frac{D_k^2}{4|E|^2}$. Now, to enforce maximal agreement, we normalize the measure as follows:

$$h_{adj} = \frac{h_{edge} - \sum_{k=1}^{C} \bar{p}(k)^2}{1 - \sum_{k=1}^{C} \bar{p}(k)^2}, \tag{2}$$

where we use the notation $\bar{p}(k) = \frac{D_k}{2|E|}$. This measure is known in graph analysis literature as *assortativity coefficient* [28]. While assortativity is a general concept that is often applied to *node degrees*, it reduces to (2) when applied to categorical node attributes on undirected graphs. Unfortunately, this measure is rarely used in graph ML literature (Suresh et al. [41] is the only work we are aware of that uses it for measuring homophily), while our theoretical analysis shows that it satisfies many desirable properties. Indeed, the following theorem holds (see Appendix B.4 for the proof).

**Theorem 1.** *Adjusted homophily satisfies maximal agreement, asymptotic constant baseline, and empty class tolerance. The minimal agreement is not satisfied. Moreover, this measure is monotone if* $h_{adj} > \frac{\sum_i \bar{p}(i)^2}{\sum_i \bar{p}(i)^2 + 1}$ *and we note that the bound* $\frac{\sum_i \bar{p}(i)^2}{\sum_i \bar{p}(i)^2 + 1}$ *is always smaller than 0.5. When* $h_{adj}$ *is small, counterexamples to monotonicity exist.*

While adjusted homophily violates some properties, it still dominates all other measures and is comparable across different datasets with varying numbers of classes and class size balance. Thus, we recommend using it as a measure of homophily in further works.

## 2.5 Edge-wise homophily vs classification evaluation measures

We conclude the analysis of homophily measures by establishing a connection between them and classification evaluation measures [8]. For this, let us first define *edge-wise* homophily measures. We say that a homophily measure is edge-wise if it is a function of the *class adjacency matrix* that we now define. Since we consider undirected graphs, each edge $\{u, v\} \in E$ gives two ordered pairs of nodes $(u, v)$ and $(v, u)$. We can define a class adjacency matrix $\mathcal{C}$ as follows: each matrix element $c_{ij}$ denotes the number of edges $(u, v)$ such that $y_u = i$ and $y_v = j$. Since the graph is undirected, the matrix $\mathcal{C}$ is symmetric. Note that monotonicity can be naturally put in terms of the class adjacency matrix: adding an edge between two nodes of the same class $i$ corresponds to incrementing the

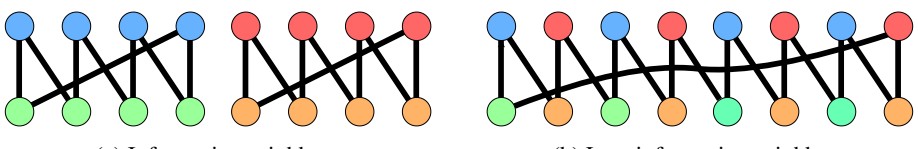

(a) Informative neighbors      (b) Less informative neighbors

Figure 1: Non-homophilous graphs with different connection patterns

diagonal element $c_{ii}$ by two, and adding an edge between two nodes of different classes $i$ and $j$ corresponds to incrementing $c_{ij}$ and $c_{ji}$ by one.

Now, for each edge $(u, v)$, let us say that $y_u$ is a *true label* (for some object) and $y_v$ is a *predicted label*. Then, any classification evaluation measure (e.g., accuracy) applied to this dataset is a measure of homophily. Based on that, we get the following correspondence.

Clearly, *accuracy* corresponds to *edge homophily* $h_{edge}$.

Interestingly, both *Cohen's Kappa* and *Matthews correlation coefficient* correspond to *adjusted homophily*. As argued in [8], the Matthews coefficient is one of the best classification evaluation measures in terms of its theoretical properties. Our extended analysis confirms this conclusion for the corresponding homophily measure: we prove a stronger version of asymptotic constant baseline and also establish a stronger variant of monotonicity in the interval of large values of $h_{adj}$. The latter result is essential since in [8] it was only claimed that monotonicity is violated when $C > 2$.

Another measure advised by [8] is *symmetric balanced accuracy*. Since in our case the class adjacency matrix is symmetric, it gives the same expression as *balanced accuracy*: $h_{bal} = \frac{1}{C} \sum_{k=1}^{C} \frac{|(u,v):y_u=y_v=k|}{D_k}$. The obtained measure satisfies the maximal and minimal agreement properties. However, it is not empty class tolerant and thus is not monotone. The asymptotic constant baseline is also not satisfied: the value $c_{base} = 1/C$ depends on the number of classes. Thus, despite this measure being suitable for classification evaluation, it cannot be used as a homophily measure. This difference is caused by the fact that homophily measures have to be comparable across datasets with different numbers of classes, while for classification evaluation it is not required.

Finally, note that similarly to our derivations in Section 2.4, the value $h_{bal}$ can be adjusted to have both maximal agreement and constant baseline. Interestingly, this would lead to a slighlty modified class homophily with $[\cdot]_+$ operation removed. As discussed in Section 2.3, the obtained measure satisfies only the maximal agreement and constant baseline.

To conclude, there is a correspondence between edge-wise homophily measures and classification evaluation measures. Adjusted homophily corresponds to both Cohen's Kappa and Matthews coefficient. In terms of the satisfied properties, adjusted homophily dominates all other measures derived from this correspondence.

Finally, we also note that homophily measures can be directly related to *community detection evaluation measures* including the well-established characteristic in graph community detection literature called *modularity* [30]. See Appendix B.5 for a detailed discussion.

## 3 Label informativeness

In the previous section, we discussed in detail how to properly measure the homophily level of a graph. While homophily indicates whether similar nodes are connected, heterophily is defined as the negation of homophily. Thus, heterophilous graphs may have very different connectivity patterns. In this section, we characterize such patterns.

To give an example, among strictly heterophilous graphs in which nodes never connect to nodes of the same class, there can be those where edges are drawn between particular pairs of classes (Figure 1a) and those where the class of a node cannot be derived from the class of its neighbor (Figure 1b). While adjusted homophily correctly captures the absence of homophily in these graphs, it is not designed to identify which type they belong to. However, distinguishing such graphs is practically important: informative neighbors can be very useful for models accounting for the graph structure.

We define a characteristic measuring the informativeness of a neighbor's label for a node's label. For example, in Figure 1a, the neighbor's label uniquely defines the node's label. Thus, the node classification task is simple on this dataset, and we want our informativeness to be maximal for such graphs. Let us formalize this idea. Assume that we sample an edge $(\xi, \eta) \in E$ (from some distribution). The class labels of nodes $\xi$ and $\eta$ are then random variables $y_\xi$ and $y_\eta$. We want to measure the amount of knowledge the label $y_\eta$ gives for predicting $y_\xi$. The entropy $H(y_\xi)$ measures the 'hardness' of predicting the label of $\xi$ without knowing $y_\eta$. Given $y_\eta$, this value is reduced to the conditional entropy $H(y_\xi|y_\eta)$. In other words, $y_\eta$ reveals $I(y_\xi, y_\eta) = H(y_\xi) - H(y_\xi|y_\eta)$ information about the label. To make the obtained quantity comparable across different datasets, we say that *label informativeness* is the normalized mutual information of $y_\xi$ and $y_\eta$:

$$\text{LI} := I(y_\xi, y_\eta)/H(y_\xi). \tag{3}$$

We have $\text{LI} \in [0, 1]$. If the label $y_\eta$ allows for unique reconstruction of $y_\xi$, then $\text{LI} = 1$. If $y_\xi$ and $y_\eta$ are independent, $\text{LI} = 0$.

Depending on the distribution used for sampling an edge $(\xi, \eta)$, one can obtain several variants of LI. For instance, if the edges are sampled uniformly at random (which is a natural approach), the mutual distribution $(y_\xi, y_\eta)$ for a randomly sampled edge is $p(c_1, c_2) = \sum_{(u,v) \in E} \frac{\mathbb{1}\{y_u = c_1, y_v = c_2\}}{2|E|}$. Then, the marginal distribution of $y_\xi$ (and $y_\eta$) is the degree-weighted distribution $\bar{p}(c)$. Thus, (3) becomes:

$$\text{LI}_{edge} = -\frac{\sum_{c_1, c_2} p(c_1, c_2) \log \frac{p(c_1, c_2)}{\bar{p}(c_1)\bar{p}(c_2)}}{\sum_c \bar{p}(c) \log \bar{p}(c)} = 2 - \frac{\sum_{c_1, c_2} p(c_1, c_2) \log p(c_1, c_2)}{\sum_c \bar{p}(c) \log \bar{p}(c)}.$$

For brevity, we further denote $\text{LI}_{edge}$ by LI and focus on this version of the measure. However, note that another natural approach to edge sampling is to first sample a random node and then sample a random edge incident to this node. For a discussion of this approach, we refer to Appendix C.2.

To claim that LI is a suitable graph characteristic, we need to show that it is comparable across different datasets. For this, we need to verify two properties: maximal agreement and asymptotic constant baseline. Recall that LI is upper bounded by one and equals one if and only if the neighbor's class uniquely reveals the node's class. This property can be considered as a direct analog of the maximal agreement defined in Section 2.2. The following theorem shows that LI satisfies the asymptotic constant baseline; see Appendix C.1 for the proof.

**Theorem 2.** *Assume that $|E| \to \infty$ as $n \to \infty$ and that the entropy of $\bar{p}(\cdot)$ is bounded from below by some constant. Let $\bar{p}_{min} = \min_k \bar{p}(k)$ and assume that $\bar{p}_{min} \gg C/\sqrt{|E|}$ as $n \to \infty$. Then, for the random configuration model, we have $\text{LI} = o(1)$ with high probability.*

In summary, LI is a simple graph characteristic suitable for comparing different datasets. We note that LI is not a measure of homophily. LI naturally complements homophily measures by distinguishing different types of heterophilous graphs.

## 4 Empirical illustrations

In this section, we first characterize some existing graph datasets in terms of homophily and LI to see which structural patterns are currently covered. Then, we show that LI, despite being a very simple graph characteristic, much better agrees with GNN performance than homophily.[5]

### 4.1 Characterizing real graph datasets

We first look at the values of homophily and label informativeness for existing graph datasets. For this analysis, we choose several node classification datasets of different sizes and properties. Statistics of these datasets and values of all the measures discussed in this paper are provided in Table 5 in Appendix E, while Table 2 shows selected results.

Recall that both node and edge homophily are sensitive to the number of classes and class size balance. Indeed, they may indicate high homophily levels for some heterophilous datasets. An

---

[5]Implementations of all the graph measures discussed in the paper and examples of their usage are provided in this Colab notebook.

extreme example is `questions`: $h_{edge} = 0.84$, while more reliable adjusted homophily shows that the dataset is heterophilous: $h_{adj} = 0.02$. In fact, all binary classification datasets in Table 5 could be considered homophilous if one chooses $h_{edge}$ or $h_{node}$ as the measure of homophily. In contrast, $h_{adj}$ satisfies the constant baseline property and shows that most of the considered binary classification datasets are heterophilous.

It is expected that datasets with high homophily $h_{adj}$ also have high LI since homophily implies informative neighbor classes. For medium-level homophily, LI can behave differently: for instance, `ogbn-arxiv` and `twitter-hate` have similar homophily levels, while the neighbors in `ogbn-arxiv` are significantly more informative. For heterophilous datasets, LI can potentially be very different, as we demonstrate in Section 4.2 on synthetic and semi-synthetic data. However, most existing heterophilous datasets have LI $\approx 0$. This issue is partially addressed by new heterophilous datasets recently proposed in [33]. For the proposed `roman-empire` dataset, LI = 0.11 and $h_{adj} = -0.05$. Thus, while being het-

Table 2: Characteristics of some real graph datasets, see Table 5 for the full results

| Dataset | $C$ | $h_{edge}$ | $h_{adj}$ | LI |
|---|---|---|---|---|
| lastfm-asia | 18 | 0.87 | 0.86 | 0.74 |
| cora | 7 | 0.81 | 0.77 | 0.59 |
| ogbn-arxiv | 40 | 0.65 | 0.59 | 0.45 |
| twitter-hate | 2 | 0.78 | 0.55 | 0.23 |
| wiki | 5 | 0.38 | 0.15 | 0.06 |
| twitch-gamers | 2 | 0.55 | 0.09 | 0.01 |
| actor | 5 | 0.22 | 0.00 | 0.00 |
| questions | 2 | 0.84 | 0.02 | 0.00 |
| roman-empire | 18 | 0.05 | -0.05 | 0.11 |

erophilous, this dataset has non-zero label informativeness, meaning that neighboring classes are somewhat informative. We believe that datasets with more interesting connectivity patterns will be collected in the future.

## 4.2 Correlation of LI with GNN performance

Recently, it has been shown that standard GNNs can sometimes perform well on non-homophilous datasets [22, 24]. We hypothesize that GNNs can learn more complex relationships between nodes than just homophily, and they will perform well as long as the node's neighbors provide some information about this node. Thus, we expect LI to better correlate with the performance of GNNs than homophily. To illustrate this, we use carefully designed synthetic and semi-synthetic data. First, it allows us to cover all combinations of homophily levels and label informativeness. Second, we can control that only a connection pattern changes while other factors affecting the performance are fixed.

**Synthetic data based on SBM model** To start with the most simple and controllable setting, we generate synthetic graphs via a variant of the *stochastic block model* (SBM) [13]. In this model, the nodes are divided into $C$ clusters, and for each pair of nodes $i, j$, we draw an edge between them with probability $p_{c(i),c(j)}$ independently of all other edges. Here $c(i)$ is a cluster assignment for a node $i$, which in our case corresponds to the node label $y_i$.

We set the number of classes to $C = 4$ and the class size to $l = n/4$. We define the probabilities as follows: $p_{i,j} = p_0 K$ if $i = j$, $p_{i,j} = p_1 K$ if $i + j = 5$, and $p_{i,j} = p_2 K$ otherwise. Here $K > 0$ and we require $p_0 + p_1 + 2p_2 = 1$. Note that the expected degree of any node is (up to a negligibly small term) $p_0 Kl + p_1 Kl + 2p_2 Kl = Kl$.

This model allows us to explore various combinations of dataset characteristics. Indeed, $p_0$ directly controls the homophily level, while the relation between $p_1$ and $p_2$ enables us to vary LI. To see this, we note that the condition $i + j = 5$ gives two pairs of classes: $(1, 4)$ and $(2, 3)$. Thus, if $p_2 = 0$ and $p_1 > 0$, knowing the label of any neighbor from another class, we can uniquely restore the node's label. In contrast, for given $p_0$, the case $p_1 = p_2$ gives the smallest amount of additional information. The following proposition characterizes the covered combinations of LI and homophily; the proof follows from the construction procedure.

**Proposition 1.** *As $n \to \infty$, the dataset characteristics of the proposed model converge to the following values (with high probability):* $h_{adj} = \frac{4}{3}p_0 - \frac{1}{3}$, LI $= 1 - \frac{H(p_0, p_1, p_2, p_2)}{\log 4}$, *where* $H(\mathrm{x}) = -\sum_i x_i \log(x_i)$.

Thus, $h_{adj}$ ranges from $-1/3$ to 1 and LI can be between 0 and 1. If LI $= 0$, then we always have $h_{adj} = 0$; if $h_{adj} = 1$, then LI $= 1$. However, if LI $= 1$, then either $h_{adj} = -1/3$ or $h_{adj} = 1$.

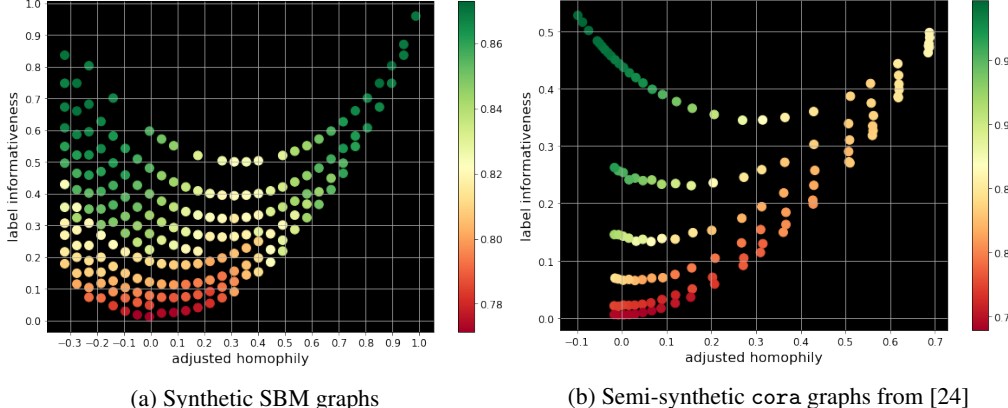

(a) Synthetic SBM graphs        (b) Semi-synthetic `cora` graphs from [24]

Figure 2: Accuracy of GraphSAGE on synthetic and semi-synthetic graphs

We generated graphs according to the procedure described above with the expected node degree of 10 and various combinations of $p_0, p_1, p_2$. Given the class labels, the features are taken from the four largest classes in the `cora` dataset [38, 27, 46, 25]. We use the obtained graphs to train four popular GNN models: **GCN** [16], **GraphSAGE** [10], **GAT** [43], and **Graph Transformer (GT)** [40]. In total, we run more than 20000 experiments on synthetic datasets for each of the four models. A detailed description of data generation, training setup, and hyperparameters used is provided in Appendix F.1.

Figure 2a shows the results for GraphSAGE. It can be seen that the performance is much more correlated with LI than with homophily. In particular, when LI is high, GraphSAGE achieves good performance even on strongly heterophilous graphs with negative homophily. We refer to Appendix F.2 for additional visualizations.

The Spearman correlation coefficient between accuracy and LI is equal to 0.93, while between accuracy and adjusted homophily it equals 0.05. For other considered models, the difference in correlation is also significant; see Table 3. Note that due to the balanced classes and degrees, all homophily measures give the same correlation, so we report only $h_{adj}$.

Table 3: Spearman correlation between model accuracy and characteristics of synthetic SBM datasets

| Model | $h_{adj}$ | $\text{LI}_{edge}$ |
|---|---|---|
| GCN | 0.19 | **0.76** |
| GraphSAGE | 0.05 | **0.93** |
| GAT | 0.17 | **0.77** |
| GT | 0.17 | **0.77** |

**Semi-synthetic data from [24]** Ma et al. [24] also argue that standard GNNs can perform well on certain heterophilous graphs. They construct semi-synthetic graphs by adding inter-class edges following different patterns to several real-world graphs, thus obtaining several sets of graphs with varying levels of homophily. Ma et al. [24] run experiments on these graphs and note that a standard GNN achieves strong performance on some heterophilous graphs.

We train GCN, GraphSAGE, GAT, and GT on the same modifications of the `cora` and `citeseer` graphs used in [24] and find that the models achieve strong performance when the graphs have high label informativeness. Training setup and hyperparameters used in these experiments are the same as above and are described in Appendix F.1. The performance of GraphSAGE on `cora` is shown in Figure 2b. In Appendix F.3, we also show the performance of GraphSAGE on the `citeseer` dataset. Table 4 shows the Spearman correlation coefficients between accuracy and various homophily measures or LI for all

Table 4: Spearman correlation between model accuracy and characteristics of semi-synthetic datasets from [24]

| Model | $h_{edge}$ | $h_{node}$ | $h_{class}$ | $h_{adj}$ | $\text{LI}_{edge}$ |
|---|---|---|---|---|---|
| | | | `cora` | | |
| GCN | -0.31 | -0.31 | -0.31 | -0.31 | **0.72** |
| GraphSAGE | -0.24 | -0.24 | -0.24 | -0.24 | **0.78** |
| GAT | -0.24 | -0.25 | -0.24 | -0.24 | **0.77** |
| GT | -0.23 | -0.24 | -0.23 | -0.23 | **0.79** |
| | | | `citeseer` | | |
| GCN | -0.24 | -0.24 | -0.24 | -0.24 | **0.76** |
| GraphSAGE | -0.53 | -0.53 | -0.54 | -0.54 | **0.51** |
| GAT | -0.27 | -0.27 | -0.27 | -0.27 | **0.75** |
| GT | -0.19 | -0.19 | -0.19 | -0.19 | **0.80** |

models and both datasets. The correlation coefficients for homophily measures are all negative, while for LI they are positive and sufficiently large. This again confirms that LI indicates whether graph structure is helpful for GNNs.

Additionally, we conduct experiments with other datasets: Appendix F.4 describes the results for the LFR benchmark [17] and Appendix F.5 presents an experiment on synthetic data from [22].

## 5   Conclusion

In this paper, we discuss how to characterize graph node classification datasets. First, we revisit the concept of homophily and show that commonly used homophily measures have significant drawbacks preventing the comparison of homophily levels between different datasets. For this, we formalize properties desirable for a good homophily measure and prove which measures satisfy which properties. Based on our analysis, we conclude that *adjusted homophily* is a better measure of homophily than the ones commonly used in the literature. We believe that being able to properly estimate the homophily level of a graph is essential for the future development of heterophily-suited GNNs: we need a characteristic that reliably differentiates homophilous and heterophilous graphs.

Then, we argue that heterophilous graphs may have very different structural patterns and propose a new property called *label informativeness* (LI) that allows one to distinguish them. LI characterizes how much information a neighbor's label provides about a node's label. Similarly to adjusted homophily, this measure satisfies important properties and thus can be used to compare datasets with different numbers of classes and class size balance. Through a series of experiments, we show that LI correlates well with the performance of GNNs.

To conclude, we believe that adjusted homophily and label informativeness will be helpful for researchers and practitioners as they allow one to easily characterize the connectivity patterns of graph datasets. We also hope that new realistic datasets will be collected to cover currently unexplored combinations of $h_{adj}$ and LI. Finally, our theoretical framework can be helpful for the further development of reliable graph characteristics.

**Limitations**   We advise using adjusted homophily as a reliable homophily measure since it is the only existing measure having constant baseline and thus it dominates other known alternatives in terms of desirable properties. However, it still violates minimal agreement, and monotonicity is guaranteed only for sufficiently large values of $h_{adj}$. It is currently an open question whether there exist measures dominating adjusted homophily in terms of the satisfied properties.

Regarding LI, we do not claim that this measure is a universal predictor of GNN performance. We designed this measure to be both informative and simple to compute and interpret. For instance, LI considers all edges individually and does not account for the node's neighborhood as a whole. As a result, LI can be insensitive to some complex dependencies. Such dependencies can be important for some tasks, but taking them into account is tricky and would significantly complicate the measure. However, we clearly see that despite its simplicity, LI correlates with GNN performance much better than homophily.

Let us also note that our analysis of both homophily and LI is limited to graph-label interactions. In future work, it would be important to also analyze node features. Indeed, node features may have non-trivial relations with both graph and labels. For example, a graph can be heterophilous in terms of labels but homophilous in terms of node features or vice versa. These interactions may allow one to understand the properties and performance of GNNs even better. However, analyzing feature-based homophily or informativeness can be much more difficult since the features can differ in nature, scale, and type.

## Acknowledgements

We thank Yao Ma for sharing the semi-synthetic datasets from Ma et al. [24] that we used in Section 4.2. We also thank Andrey Ploskonosov for thoughtful discussions.

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

# A  Random configuration model

Numerous random graph models have been proposed to reflect and predict important quantitative and topological aspects of real-world networks [3]. The simplest model is the Erdős–Rényi random graph, i.e., we assume that $G$ is sampled uniformly at random from the set of all graphs with $n$ nodes and $|E|$ edges. However, this model is known to be not a good descriptor of real-world networks since its Poisson degree distribution significantly differs from heavy-tailed degree distributions observed in real-world networks. A standard solution is to consider a random graph with a given degree sequence [26]: a graph is sampled uniformly from the set of all graphs with a given degree sequence.

A *configuration model*, which we assume throughout the paper, is defined as follows. To generate a graph, we form a set $A$ containing $d(v)$ distinct copies of each node $v$ and then choose a random matching of the elements of $A$. In this case, self-loops and multiple edges may appear. Under some conditions, the obtained graph is simple (i.e., does not contain self-loops and multiple edges) with probability $1 - o(1)$ [26].

Let us also note that there is another model that is similar to the one discussed above but can be easier to analyze. To obtain *a graph with given expected degree sequence* [5], we take the degree sequence from the observed graph and say that the number of edges between $i$ and $j$ follows a Poisson distribution with the mean $\frac{d(i)d(j)}{2|E|}$ if $i \neq i$ and the expected number of self-loops for a node $i$ is $\frac{d(i)^2}{4|E|}$. This model does not preserve the exact degree sequence but has it in expectation. Note that usually $d(i)d(j) \ll 2|E|$, so multiple edges rarely appear. Asymptotically, this model is similar to the configuration model, but a graph with a given expected degree sequence is easier to analyze theoretically. In our analysis, we assume the configuration model, so we have to track the error terms carefully.

# B  Analysis of homophily

## B.1  Class homophily

Recall that class homophily is defined as [20]:

$$h_{class} = \frac{1}{C-1} \sum_{k=1}^{C} \left[ \frac{\sum_{v:y_v=k} |\{u \in N(v) : y_u = y_v\}|}{\sum_{v:y_v=k} d(v)} - \frac{n_k}{n} \right]_+ .$$

Here the first term inside the brackets is the fraction of edges that go from a particular class $k$ to itself. The second term $n_k/n = p(k)$ is the null expected fraction. We note that $p(k)$ is the expected fraction if we assume the Erdős–Rényi model. Indeed, for this model, all edges have equal probability, and thus the expected fraction of neighbors of class $k$ is proportional to the size of this class. However, as mentioned above, the Erdős–Rényi null model has certain disadvantages since it does not take into account node degrees. This may lead to incorrect estimates of null probabilities for degree-imbalanced classes, as shown in the example below.[6]

**Proposition 2.** *Assume that we have two classes of size $n/2$. Further, assume that the expected degrees of nodes in the first class are equal to $d$, while nodes in the second class have expected degrees $ld$ for some $l > 1$. Then, if edges are added independently of the classes, the expected value of $h_{class}$ is*

$$\mathbb{E}h_{class} = \frac{l}{l+1} - \frac{1}{2}.$$

*Thus, for randomly connected nodes we get $\mathbb{E}h_{class} \to 1/2$ as $l \to \infty$.*

*Proof.* We need to compute the expected number of intra-class edges for each class. For the first class, we multiply the number of nodes by the degree of each node and by the probability of a particular

---

[6]For simplicity, in these statements, we use the random graph model with a given expected degree sequence as it allows for simpler illustrations. For the configuration model, the error terms can be tracked similarly to the proofs in Sections B.4 and C.1.

edge to go to the same class: $\frac{n}{2} \cdot d \cdot \frac{nd}{n(l+1)d} = \frac{nd}{2(l+1)}$. Normalizing by the sum of the degrees, we get $\frac{1}{(l+1)}$. Similarly, for the second class, the normalized number of the intra-class edges is $\frac{l}{(l+1)}$. Hence,

$$\mathbb{E}h_{class} = \left[\frac{1}{l+1} - \frac{1}{2}\right]_+ + \left[\frac{l}{l+1} - \frac{1}{2}\right]_+ = \frac{l}{l+1} - \frac{1}{2}.$$

$\square$

This proposition shows that class homophily does not satisfy the constant baseline property.

As discussed in Section 2.3, if we remove the $[\cdot]_+$ operation from the definition of class homophily, we obtain a measure that does satisfy the asymptotic constant baseline. Following Section 2.5, we call this measure *balanced adjusted homophily* as it adjusts balance homophily for chance:

$$h_{bal}^{adj} = \frac{1}{C-1} \sum_{k=1}^{C} \left( \frac{\sum_{v:y_v=k} |\{u \in N(v) : y_u = y_v\}|}{\sum_{v:y_v=k} d(v)} - \frac{n_k}{n} \right) = \frac{Ch_{bal} - 1}{C - 1}.$$

**Proposition 3.** *Assuming that there are $C$ classes and the graph is generated according to the model with given expected degrees, we have $\mathbb{E}h_{bal}^{adj} = 0$.*

*Proof.* Let us denote the sizes of the classes by $n_1, \ldots, n_C$. Recall that we use the notation $D_k = \sum_{v:y_v=k} d(v)$. It is easy to see that

$$\mathbb{E}h_{bal}^{adj} = \frac{1}{C-1} \sum_{i=1}^{C} \left( \frac{D_i}{\sum_j D_j} - \frac{n_i}{\sum_j n_j} \right) = \frac{1-1}{C-1} = 0.$$

$\square$

## B.2 Constant baseline

There are several possible ways to formalize the constant baseline property. In the literature [8, 9], this property is often formalized as follows: assuming some randomized model, the expected value of a measure should be equal to some constant. For homophily measures, this corresponds to the following definition.

**Definition 5.** A homophily measure $h$ has *constant baseline* if for $G$ generated according to the configuration model we have $\mathbb{E}h(G) = c_{base}$ for some constant $c_{base}$.

This property is very strict as minor variations in the definition of the model (e.g., different alternatives of the configuration model discussed in Appendix A) may lead to negligibly small error terms preventing us from getting exactly the same constant $c_{base}$. Thus, in the main text, we use the alternative Definition 2. This definition is weaker in the sense that we allow asymptotically negligible deviations from the constant $c_{base}$. On the other hand, our definition is somewhat stronger since we also require the concentration of the value around its expectation. In that sense, our definition is stronger than the *asymptotic* constant baseline defined in [8].

## B.3 Monotonicity property

In this section, we discuss how one can define monotonicity for homophily measures.

First, let us revisit how monotonicity is defined for measures used in other areas. In general, the monotonicity property is defined as follows: if an object $B'$ is definitely more similar to an object $A$ than $B$, then the similarity between $B'$ and $A$ should be larger than between $B$ and $A$. The only problem is to define what it means to be "definitely more similar". In [9], the monotonicity property was introduced for cluster similarity measures. Such measures evaluate how close two partitions of the same set of items are. To formally define the concept of more similar partitions, Gösgens et al. [9] use the concept of perfect splits and perfect merges — such transformations of a partition $B$ that make it more similar to $A$. Later, monotonicity was defined for classification evaluation measures that compare the predicted labels for a set of items with the reference ones [8]. Here it is easier to

formalize what it means that a labeling $B'$ is definitely more similar to the reference labeling than $B$. Indeed, if we consider a labeling $B$ and change one incorrect label to the correct one, then it becomes closer to the reference labeling.

Now, let us return to homophily measures. Such measures evaluate the agreement between *a graph* and *a labeling* of its nodes. To define monotonicity, we need such transformations that make a graph and a labeling definitely more similar. Since the problem is not symmetric, we can either rewire edges or relabel nodes. Regarding relabeling of nodes, we may say that taking a perfectly heterophilous node (that is not connected to any node of its class) and relabeling it in such a way that it becomes perfectly homophilous (which is only possible when all its neighbors belong to one class) should make the graph and labeling more similar to each other. Regarding the edges, we may say that taking an edge that connects two nodes of different classes and rewiring it to connect two nodes of the same class should increase the measure. However, rewiring edges may also affect the graph structure in a non-trivial way (degree distribution, diameter, etc.).

In this paper, we follow the edge rewiring approach, and thus our definition aligns well with edge-wise homophily measures and with previous works on classification and clustering performance measures [8, 9].

If we consider edge-wise measures (that can be defined in terms of the class adjacency matrix) and directly follow the definition of monotonicity in [8, 9], our definition would be as follows: a homophily measure is *monotone* if it increases when we decrement $c_{ij}$ and $c_{ji}$ by one and increment either $c_{ii}$ or $c_{jj}$ by two for $i \neq j$. This condition corresponds to taking an edge with $y_u \neq y_v$ and changing $y_u$ to $y_v$ or vice versa.

However, an important property that has to be taken into account when discussing monotonicity for homophily measures is the fact that such measures are used to compare different graph datasets, in contrast to classification evaluation measures that compare the agreement between predicted labelings with a *fixed* given reference labeling. This means that for classification measures, monotonicity and constant baseline are critical for fixed numbers of classes and elements. In contrast, homophily measures have to be comparable across datasets with different sizes and numbers of classes. For instance, balanced accuracy has a constant baseline with the expected value of $\frac{1}{C}$, which is sufficient for classification evaluation but is a drawback for measuring homophily. Because of this, we do not consider the standard definition of monotonicity as it is restricted to a fixed number of classes only. Instead, we introduce tolerance to empty classes, and based on that, we introduce a stronger version of monotonicity where we allow separate edge additions or deletions. In fact, our definition of monotonicity (Definition 4) is similar to *strong monotonicity* in [8, 9], but also requires empty class tolerance since comparing measures across different numbers of classes is crucial.

## B.4 Proof of Theorem 1

The fact that adjusted homophily is empty class tolerant directly follows from its definition: an empty class does not contribute to the numerator or denominator of $h_{adj}$.

Maximal agreement is also straightforward: we have $h_{edge} - \sum_{k=1}^{C} D_k^2/(2|E|)^2 \leq 1 - \sum_{k=1}^{C} D_k^2/(2|E|)^2$ with equality if and only if all edges are homophilous.

Minimal agreement is not satisfied since the value $\frac{-\sum_{k=1}^{C} D_k^2/(2|E|)^2}{1-\sum_{k=1}^{C} D_k^2/(2|E|)^2}$ can be different for different datasets. Thus, perfectly heterophilous datasets may get different values, which also causes some monotonicity violations for small values of $h_{adj}$.

**Asymptotic constant baseline of adjusted homophily**    Now, let us formulate and prove the constant baseline property.

**Proposition 4.** *Let $G$ be a graph generated according to the configuration model. Assume that the degree-weighted distribution of classes is not very unbalanced, i.e., that $1 - \sum_k \bar{p}(k)^2 \gg 1/\sqrt{E}$. Then, $|h_{adj}| \leq \phi$ with probability $1 - o(1)$ for some $\phi = \phi(|E|) \to 0$ as $|E| \to \infty$.*

*Proof.* Let us first analyze the numerator of $h_{adj}$ which we denote by $h_{mod}$ (since it corresponds to the network's *modularity* — see Appendix B.5).

Let $\xi$ denote the number of intra-class edges in a graph constructed according to the configuration model. We may write:

$$\xi = \frac{1}{2} \sum_{k=1}^{C} \sum_{i=1}^{D_k} \mathbb{1}_{i,k} \,,$$

where $\mathbb{1}_{i,k}$ indicates that an endpoint $i$ of some node with class $k$ is connected to another endpoint of class $k$. Note that the probability of this event is $\frac{D_k-1}{2|E|-1}$. Thus, we have

$$\mathbb{E}h_{mod} = \frac{\xi}{|E|} - \sum_{k=1}^{C} \frac{D_k^2}{4|E|^2} = \frac{1}{|E|} \sum_{k=1}^{C} \frac{D_k(D_k-1)}{2(2|E|-1)} - \sum_{k=1}^{C} \frac{D_k^2}{4|E|^2} = O\left(\frac{1}{|E|}\right). \qquad (4)$$

Now, let us estimate the variance of the number of intra-class edges. We may write:

$$\mathrm{Var}(2\xi) = \mathbb{E}(2\xi)^2 - (\mathbb{E}(2\xi))^2 = \mathbb{E}\left(\sum_{k=1}^{C} \sum_{i=1}^{D_k} \mathbb{1}_{i,k}\right)^2 - (\mathbb{E}2\xi)^2$$

$$= \mathbb{E}(2\xi) + \sum_{k=1}^{C} 2 \cdot \binom{D_k}{2} \cdot \left(\frac{1}{2|E|-1} + \frac{D_k-2}{2|E|-2} \cdot \frac{D_k-3}{2|E|-3}\right)$$

$$+ \sum_{k=1}^{C} \sum_{l=k+1}^{C} 2D_k D_l \cdot \frac{D_k-1}{2|E|-1} \cdot \frac{D_l-1}{2|E|-3} - (\mathbb{E}2\xi)^2$$

$$= \mathbb{E}(2\xi) + \sum_{k=1}^{C} \frac{D_k^4 + O(D_k^3)}{4|E|^2}$$

$$+ \sum_{k=1}^{C} \sum_{l=k+1}^{C} \frac{2D_k^2 D_l^2 + O(D_k^2 D_l + D_k D_l^2)}{4|E|^2} - \left(\sum_{k=1}^{C} \frac{D_k^2 + O(D_k)}{2|E|}\right)^2$$

$$= \mathbb{E}(2\xi) + \sum_{k=1}^{C} \frac{O(D_k^3)}{4|E|^2} + \sum_{k=1}^{C} \sum_{l=k+1}^{C} \frac{O(D_k^2 D_l + D_k D_l^2)}{4|E|^2}$$

$$= \mathbb{E}(2\xi) + O(|E|) = O(|E|) \,.$$

Let $\varphi = \varphi(|E|)$ be any function such that $\varphi \to \infty$ as $|E| \to \infty$. Using the Chebyshev's inequality and (4), we get:

$$\mathrm{P}\left(|h_{mod}| \geq \frac{\varphi}{\sqrt{E}}\right) = \mathrm{P}\left(|h_{mod} - \mathbb{E}h_{mod}| > \frac{\varphi}{\sqrt{E}} + O\left(\frac{1}{|E|}\right)\right)$$

$$= O\left(\frac{\mathrm{Var}(\xi)|E|}{|E|^2 \varphi^2}\right) = O\left(\frac{1}{\varphi^2}\right) = o(1).$$

Recall that $h_{adj} = \frac{h_{mod}}{1 - \sum_{k=1}^{C} \bar{p}(k)^2}$. Since we have $1 - \sum_{k=1}^{C} \bar{p}(k)^2 \gg 1/\sqrt{E}$ and $|h_{mod}| < \frac{\varphi}{\sqrt{E}}$ with probability $1 - o(1)$, we can choose such slowly growing $\varphi$ that $h_{adj} < \frac{\varphi}{\sqrt{E}\left(1 - \sum_{k=1}^{C} \bar{p}(k)^2\right)} = o(1)$ with probability $1 - o(1)$.

$\square$

**Monotonicity of adjusted homophily**  Finally, let us analyze the monotonicity of adjusted homophily and finish the proof of Theorem 1.

Recall that a homophily measure is *monotone* if it is empty class tolerant, and it increases when we increment a diagonal element by 2 (except for perfectly homophilous graphs) and decreases when we increase $c_{ij}$ and $c_{ji}$ by one for $i \neq j$ (except for perfectly heterophilous graphs).

Empty class tolerance is clearly satisfied for adjusted homophily, so let us now analyze what happens when we increment a diagonal element or two (symmetric) off-diagonal elements.

Let us denote $N := 2|E|$, $a_i := \sum_j c_{ij}$. Then, we have $\bar{p}(i) = \frac{a_i}{N}$. Thus, we have to prove that the measure is monotone when $h_{adj} > \frac{\sum_i a_i^2}{(\sum_i a_i^2 + n^2)}$.

Using the notation with a class adjacency matrix, adjusted homophily can be written as follows:

$$h_{adj} = \frac{N \sum_i c_{ii} - \sum_i a_i^2}{N^2 - \sum_i a_i^2}.$$

To check whether the measure increases when we increment diagonal elements, let us compute the derivative w.r.t. $c_{kk}$ for some $k$:

$$\frac{\partial h_{adj}}{\partial c_{kk}} = \frac{\left(\sum_i c_{ii} + N - 2a_k\right)\left(N^2 - \sum_i a_i^2\right)}{\left(N^2 - \sum_i a_i^2\right)^2} - \frac{(2N - 2a_k)\left(N \sum_i c_{ii} - \sum_i a_i^2\right)}{\left(N^2 - \sum_i a_i^2\right)^2}.$$

Let us simplify the numerator:

$$\left(\sum_i c_{ii} + N - 2a_k\right)\left(N^2 - \sum_i a_i^2\right) - (2N - 2a_k)\left(N \sum_i c_{ii} - \sum_i a_i^2\right)$$
$$= N^3 - 2N^2 a_k - \sum_i c_{ii} \sum_i a_i^2 - N^2 \sum_i c_{ii} + 2a_k N \sum_i c_{ii} + N \sum_i a_i^2.$$

For monotonicity, we need the derivative to be positive, i.e.,

$$N^3 + 2a_k N \sum_i c_{ii} + N \sum_i a_i^2 > 2N^2 a_k + \sum_i c_{ii} \sum_i a_i^2 + N^2 \sum_i c_{ii}.$$

Let us denote by $\bar{D}$ the sum of all off-diagonal elements, i.e., $\bar{D} := N - \sum_i c_{ii}$. Then, we can rewrite the above condition as follows:

$$N^2 \bar{D} + \bar{D} \sum_i a_i^2 > 2N a_k \bar{D},$$

$$N^2 + \sum_i a_i^2 > 2N a_k.$$

The latter equality holds since

$$N^2 + \sum_i a_i^2 > N^2 + a_k^2 \geq 2N a_k.$$

Thus, $h_{adj}$ increases when we increment a diagonal element.

To see whether the measure decreases when we increment the off-diagonal elements, let us compute the derivative of the measure w.r.t. $c_{kl}$ (which is equal to $c_{lk}$) for $k \neq l$:

$$\frac{\partial h_{adj}}{\partial c_{kl}} = \frac{\left(2\sum_i c_{ii} - 2a_k - 2a_l\right)\left(N^2 - \sum_i a_i^2\right)}{\left(N^2 - \sum_i a_i^2\right)^2} - \frac{(4N - 2a_k - 2a_l)\left(N \sum_i c_{ii} - \sum_i a_i^2\right)}{\left(N^2 - \sum_i a_i^2\right)^2}.$$

The numerator is:

$$\left(2\sum_i c_{ii} - 2a_k - 2a_l\right)\left(N^2 - \sum_i a_i^2\right) - (4N - 2a_k - 2a_l)\left(N \sum_i c_{ii} - \sum_i a_i^2\right)$$
$$= -2(a_k + a_l)N^2 - 2\sum_i c_{ii} \sum_i a_i^2 - 2N^2 \sum_i c_{ii} + 2(a_k + a_l)N \sum_i c_{ii} + 4N \sum_i a_i^2.$$

For monotonicity, we need the following inequality:

$$(a_k + a_l)N^2 + \sum_i c_{ii} \sum_i a_i^2 + N^2 \sum_i c_{ii} > (a_k + a_l)N \sum_i c_{ii} + 2N \sum_i a_i^2. \qquad (5)$$

This inequality is not always satisfied. Indeed, both $a_k + a_l$ and $\sum_i c_{ii}$ can be small. Let us note, however, that the inequality is satisfied if $h_{adj}$ is large enough. Indeed, the sufficient condition for (5) is

$$\sum_i c_{ii}\left(\sum_i a_i^2 + N^2\right) > 2N \sum_i a_i^2.$$

This is equivalent to the following inequality for $h_{adj}$:

$$h_{adj} = \frac{N\sum_i c_{ii} - \sum_i a_i^2}{N^2 - \sum_i a_i^2} > \frac{N\sum_i \frac{2N\sum_i a_i^2}{\left(\sum_i a_i^2 + N^2\right)} - \sum_i a_i^2}{N^2 - \sum_i a_i^2}$$

$$= \frac{N^2\sum_i a_i^2 - \left(\sum_i a_i^2\right)^2}{\left(N^2 - \sum_i a_i^2\right)\left(\sum_i a_i^2 + N^2\right)} = \frac{\sum_i a_i^2}{\left(\sum_i a_i^2 + N^2\right)} .$$

Thus, $h_{adj}$ is monotone if its values are at least $\frac{\sum_i a_i^2}{\left(\sum_i a_i^2 + N^2\right)}$. Note that we have $\frac{\sum_i a_i^2}{\left(\sum_i a_i^2 + N^2\right)} < 0.5$, so when $h_{adj} > 0.5$ it is always monotone.

However, non-monotone behavior may occur when the numerator of $h_{adj}$ is small. This undesirable behavior is somewhat expected: due to the normalization, $h_{adj}$ satisfies constant baseline and maximal agreement but violates minimal agreement. Since minimal agreement does not hold, one can expect monotonicity to be violated for small values of $h_{adj}$.

For instance, we can construct the following counter-example to monotonicity. Assume that we have four classes $(0, 1, 2, 3)$ and non-zero entries of the class adjacency matrix are $c_{23} = c_{32} = M$, $c_{33} = 2$. Then, the adjusted homophily is:

$$h_{adj} = \frac{2(2M+2) - M^2 - (M+2)^2}{(2M+2)^2 - M^2 - (M+2)^2} = \frac{-2M^2}{2M^2 + 4M} = \frac{-M}{M+2} .$$

Now, we increment the entries $c_{01}$ and $c_{10}$ by 1. The new adjusted homophily is:

$$h'_{adj} = \frac{2(2M+4) - M^2 - (M+2)^2 - 2}{(2M+4)^2 - M^2 - (M+2)^2 - 2} = \frac{-2M^2 + 2}{2M^2 + 12M + 10} = \frac{1-M}{M+5} .$$

We disprove monotonicity if we have $h'_{adj} > h_{adj}$, i.e.,

$$\frac{1-M}{M+5} > \frac{-M}{M+2} ,$$
$$2 - M - M^2 > -M^2 - 5M ,$$
$$2 > -4M ,$$

which holds for all $M \geq 1$.

## B.5 How homophily and modularity are related

**Modularity**  *Modularity* is arguably the most well-known measure of goodness of a partition for a graph. It was first introduced in [30] and is widely used in community detection literature: modularity is directly optimized by some algorithms, used as a stopping criterion in iterative methods, or used as a metric to compare different algorithms when no ground truth partition is available. The basic idea is to consider the fraction of intra-community edges among all edges of $G$ and penalize it for avoiding trivial partitions like those consisting of only one community of size $n$. In its general form and using the notation adopted in this paper, modularity is

$$\frac{1}{|E|}\left(|\{\{u,v\} \in E : y_u = y_v\}| - \gamma\mathbb{E}\xi\right) ,$$

where $\xi$ is a random number of intra-class edges in a graph constructed according to some underlying random graph model; $\gamma$ is the *resolution parameter* which allows for varying the number of communities obtained after maximizing modularity. The standard choice is $\gamma = 1$, which also guarantees that the expected value of modularity is 0 if a graph is generated independently of class labels according to the underlying model.

Usually, modularity assumes the configuration model. In this case, we have $\mathbb{E}\xi = \frac{\sum_k D_k(D_k-1)}{2(2|E|-1)} \approx \frac{1}{4|E|}\sum_k D_k^2$, giving the following expression:

$$h_{mod} = \frac{|\{\{u,v\} \in E : y_u = y_v\}|}{|E|} - \sum_{k=1}^{C} \frac{D_k^2}{4|E|^2} .$$

We refer to Newman [29], Prokhorenkova and Tikhonov [34] for more details regarding modularity and its usage in community detection literature.

**Relation to homophily**  While modularity measures how well a partition fits a given graph, homophily measures how well graph edges agree with the partition (class labels). Thus, they essentially measure the same thing, and modularity can be used as a homophily measure. Indeed, it is easy to see that modularity coincides with the numerator of adjusted homophily, see (2). Hence, adjusted homophily can be viewed as a normalized version of modularity. Note that for modularity, normalization is not crucial as modularity is usually used to compare several partitions of the same graph. In contrast, homophily is typically used to compare different graphs, which is why normalization is essential.

## C  Analysis of label informativeness

### C.1  Proof of Theorem 2

In this section, we prove Theorem 2. Let us give a formal statement of this theorem.

**Theorem 2.** *Assume that $|E| \to \infty$ as $n \to \infty$. Assume that the entropy of $\bar{p}(\cdot)$ is bounded from below by some constant. Let $\bar{p}_{min} = \min_k \bar{p}(k)$. Assume that $\bar{p}_{min} \gg C/\sqrt{|E|}$ as $n \to \infty$. Let $\varepsilon = \varepsilon(n)$ be any function such that $\varepsilon \gg \dfrac{C}{p_{min}\sqrt{|E|}}$ and $\varepsilon = o(1)$ as $n \to \infty$. Then, with probability $1 - o(1)$, we have $|\mathrm{LI}| \le K\varepsilon = o(1)$ for some constant $K > 0$.*

*Proof.* Recall that

$$\mathrm{LI} = -\frac{\sum_{c_1,c_2} p(c_1,c_2) \log \frac{p(c_1,c_2)}{\bar{p}(c_1)\bar{p}(c_2)}}{\sum_c \bar{p}(c) \log \bar{p}(c)}.$$

Thus, for each pair $k, l$, we need to estimated $p(k, l)$. Let us denote:

$$E(k,l) := \sum_{u=1}^{n} \sum_{v=1}^{n} \mathbb{1}\{\{u,v\} \in E, y_u = k, y_v = l\},$$

so we have $p(k,l) = \frac{E(k,l)}{2|E|}$.

As before, we use the notation $D_k = \sum_{v:y_v=k} d(v)$. First, consider the case $k \ne l$. Let us compute the expectation of $E(k,l)$:

$$\mathbb{E}E(k,l) = \frac{D_k D_l}{2|E| - 1} \text{ for } k \ne l,$$

$$\mathbb{E}E(k,k) = \frac{D_k(D_k - 1)}{2|E| - 1}.$$

Now, we estimate the variance. Below, $\mathbb{1}_{i,j}$ indicates that two endpoints are connected.

$$\mathrm{Var}E(k,l) = \mathbb{E}\left(\sum_{i=1}^{D_k} \sum_{j=1}^{D_l} \mathbb{1}_{i,j}\right)^2 - (\mathbb{E}E(k,l))^2$$

$$= \mathbb{E}E(k,l) + \frac{D_k D_l (D_k - 1)(D_l - 1)}{(2|E| - 1)(2|E| - 3)} - (\mathbb{E}E(k,l))^2$$

$$= \mathbb{E}E(k,l) + O\left(\frac{D_k D_l (D_k + D_l)}{|E|^2}\right) = O\left(\frac{D_k D_l}{|E|}\right).$$

Similarly,

$$\mathrm{Var}E(k,k) = \mathbb{E}\left(\sum_{i=1}^{D_k} \sum_{j=1}^{D_k} \mathbb{1}_{i,j}\right)^2 - (\mathbb{E}E(k,k))^2 = \mathbb{E}E(k,k) + O\left(\frac{D_k^3}{|E|^2}\right) = O\left(\frac{D_k^3}{|E|}\right).$$

From Chebyshev's inequality, we get:

$$\mathrm{P}\left(|E(k,l) - \mathbb{E}E(k,l)| > \varepsilon \mathbb{E}E(k,l)\right) = O\left(\frac{|E|}{D_k D_l \varepsilon^2}\right) = O\left(\frac{1}{\bar{p}_{min}^2 |E| \varepsilon^2}\right) = o\left(\frac{1}{C^2}\right).$$

Thus, with probability $1 - o(1)$, $\mathrm{P}\left(|E(k,l) - \mathbb{E}E(k,l)| < \varepsilon \mathbb{E}E(k,l)\right)$ for all pairs of classes. In this case,

$$\mathrm{LI} = -\frac{\sum_{k,l} p(k,l) \log(1 + O(\varepsilon))}{\sum_l \bar{p}(k) \log \bar{p}(k)} = O(\varepsilon) \,.$$

$\square$

### C.2 Alternative definition

Recall that in Section 3 we define the label informativeness in the following general form: $\mathrm{LI} := I(y_\xi, y_\eta)/H(y_\xi)$. Then, to define $\mathrm{LI}_{edge}$, we say that $\xi$ and $\eta$ are two endpoints of an edge sampled uniformly at random. Another possible variant is when we first sample a random node and then sample its random neighbor. The probability of an edge becomes

$$\bar{p}(c_1, c_2) = \sum_{(u,v) \in E} \frac{\mathbb{1}\{y_u = c_1, y_v = c_2\}}{n\,d(u)}\,.$$

In this case, $H(y_\xi)$ is the entropy of the distribution $p(c)$, $H(y_\eta)$ is the entropy of $\bar{p}(c)$. Thus, we obtain:

$$\mathrm{LI}_{node} = -\frac{\sum_{c_1, c_2} \bar{p}(c_1, c_2) \log \frac{\bar{p}(c_1, c_2)}{p(c_1)\bar{p}(c_2)}}{\sum_c p(c) \log p(c)}\,.$$

In this paper, we mainly focus on $\mathrm{LI}_{edge}$ and refer to it as LI for brevity. First, this measure is conceptually similar to adjusted homophily discussed above: they both give equal weights to all edges. Second, in our analysis of real datasets, we do not notice a substantial difference between $\mathrm{LI}_{edge}$ and $\mathrm{LI}_{node}$ in most of the cases, see Table 5. However, these measures can potentially be different, especially for graphs with very unbalanced degree distributions (in $\mathrm{LI}_{node}$, averaging is over the nodes, so all nodes are weighted equally, while in $\mathrm{LI}_{edge}$, averaging is over the edges, so more weight is given to high-degree nodes). The choice between $\mathrm{LI}_{edge}$ and $\mathrm{LI}_{node}$ may depend on a particular application.

## D   Additional related work

In the aspect of characterizing graph datasets, our work is conceptually similar to a recent paper by Liu et al. [21]. In this paper, the authors empirically analyze what aspects of a graph dataset (e.g., node features or graph structure) influence the GNN performance. Liu et al. [21] follow a data-driven approach and measure the performance change caused by several data perturbations. In contrast, we follow a theoretical approach to choosing graph characteristics based only on the label-structure relation. Additionally, our characteristics are very simple, model agnostic, and can be used for general-purpose graph analysis (beyond graph ML). Similarly to [21], we believe that the proposed characteristics can help in the selection and development of diverse future graph benchmarks.

Similarly to our work, several papers note that homophily does not always reflect the simplicity of a dataset for GNNs, and standard GNNs can work well on some heterophilous graphs. To address this problem, Luan et al. [22] propose a metric called *aggregation homophily* that takes into account both graph structure and input features. We note that the aggregation homophily is based on a particular aggregation scheme of the GCN model. In contrast, the proposed LI is a simple and intuitive model-agnostic measure. An additional advantage of LI is that it is provably unbiased and can be compared across datasets with different numbers of classes and class size balance. Another feature-based metric was very recently proposed by Luan et al. [23]. It is called Kernel Performance Metric (KPM) and reflects how beneficial is a graph structure for prediction. Ma et al. [24] also observe that some heterophilous graphs are easy for GNNs. To analyze this problem, they propose a measure named *cross-class neighborhood similarity* defined for pairs of classes. While this measure is an informative tool to analyze a particular dataset, it does not give a single number to easily compare different datasets.

Finally, let us note that Suresh et al. [41] use (2) as a measure of graph assortativity (homophily). The authors show that the homophily level varies over the graph, and the prediction performance of GNNs correlates with the local homophily. They use these insights to transform the input graph and

get an enhanced level of homophily. In future work, it would be interesting to see whether additional benefits can be obtained using label informativeness instead of homophily.

# E  Characterizing real graph datasets

## E.1  Datasets

`Cora`, `citeseer`, and `pubmed` [6, 25, 38, 27, 46] are three classic paper citation network benchmarks. For `cora` and `citeseer` labels correspond to paper topics, while for `pubmed` labels specify the type of diabetes addressed in the paper. `Coauthor-cs` and `coauthor-physics` [39] are co-authorship networks. Nodes represent authors, and two nodes are connected by an edge if the authors co-authored a paper. Node labels correspond to fields of study. `Amazon-computers` and `amazon-photo` [39] are co-purchasing networks. Nodes represent products, and an edge means that two products are frequently bought together. Labels correspond to product categories. `Lastfm-asia` is a social network of music streaming site LastFM users who live in Asia [35]. Edges represent follower relationships, and labels correspond to user's nationality. In the `facebook` [37] graph nodes correspond to official Facebook pages, and links indicate mutual likes. Labels represent site categories. In the `github` [37] graph, nodes represent GitHub users and edges represent follower relationships. A binary label indicates that a user is either a web or a machine learning developer. `Ogbn-arxiv` and `Ogbn-products` [14] are two datasets from the recently proposed Open Graph Benchmark. `Ogbn-arxiv` is a citation network graph with labels corresponding to subject areas. `ogbn-products` is a co-purchasing network with labels corresponding to product categories. `Actor` [42, 32] is a popular dataset for node classification in heterophilous graphs. The nodes correspond to actors and edges represent co-occurrence on the same Wikipedia page. The labels are based on words from an actor's Wikipedia page. `Flickr` [47] is a graph of images with labels corresponding to image types. `Deezer-europe` [35] is a user network of the music streaming service Deezer with labels representing a user's gender. `Twitch-de` and `twitch-pt` [37] are social network graphs of gamers from the streaming service Twitch.[7] The labels indicate if a streamer uses explicit language. `Genius` [19], `twitch-gamers` [36], `arxiv-year` [14], `snap-patents` [18], and `wiki` [20] are recently proposed large-scale heterophilous datasets. For the `wiki` dataset we remove all isolated nodes. `Roman-empire`, `amazon-ratings`, `minesweeper`, `workers`, and `questions` [33] are recently proposed mid-scale heterophilous datasets. We additionally construct one more binary classification graph — `twitter-hate`. This graph is based on data from Hateful Users on Twitter dataset on Kaggle.[8] The labels indicate if a user posts hateful messages or not. We remove all the unlabeled nodes from the graph and use the largest connected component of the resulting graph.

We transform all the considered graphs to simple undirected graphs and remove self-loops.

## E.2  Dataset characteristics

Dataset characteristics are shown in Table 5. This table extends Table 2 in the main text. It can be seen that the typically used homophily measures — $h_{edge}$ and $h_{node}$ — often overestimate homophily levels, since they do not take into account the number of classes and class size balance. This is particularly noticeable for datasets with two classes. If fact, according to these measures all binary classification datasets in our table are homophilous. In contrast, $h_{adj}$ corrects for the expected number of edges between classes and shows that most of the considered binary classification datasets are actually heterophilous (`github` and `twitter-hate` being the exceptions).

As for label informativeness, it can be seen that on real heterophilous datasets it is typically very low (with the exception of `roman-empire` dataset). This is in contrast to synthetic datasets used for experiments in [22, 24], which sometimes exhibit a combination of low homophily and high label informativeness. High label informativeness of these datasets leads to strong GNN performance on them despite low homophily levels.

Let us note that edge and node label informativeness differ in how they weight high/low-degree nodes. For node label informativeness, averaging is over the nodes, so all nodes are weighted equally. For

---

[7]`Twitch-de` and `twitch-pt` are subgraphs of a larger dataset `twitch-gamers`. We report characteristics for all of them since they have different sizes, edge density, and may have different structural properties.

[8]https://www.kaggle.com/datasets/manoelribeiro/hateful-users-on-twitter

Table 5: Dataset characteristics, more homophilous datasets are above the line

| Dataset | $n$ | $|E|$ | $C$ | $h_{edge}$ | $h_{node}$ | $h_{class}$ | $h_{adj}$ | $\text{LI}_{edge}$ | $\text{LI}_{node}$ |
|---|---|---|---|---|---|---|---|---|---|
| cora | 2708 | 5278 | 7 | 0.81 | 0.83 | 0.77 | 0.77 | 0.59 | 0.61 |
| citeseer | 3327 | 4552 | 6 | 0.74 | 0.72 | 0.63 | 0.67 | 0.45 | 0.45 |
| pubmed | 19717 | 44324 | 3 | 0.80 | 0.79 | 0.66 | 0.69 | 0.41 | 0.40 |
| coauthor-cs | 18333 | 81894 | 15 | 0.81 | 0.83 | 0.75 | 0.78 | 0.65 | 0.68 |
| coauthor-physics | 34493 | 247962 | 5 | 0.93 | 0.92 | 0.85 | 0.87 | 0.72 | 0.76 |
| amazon-computers | 13752 | 245861 | 10 | 0.78 | 0.80 | 0.70 | 0.68 | 0.53 | 0.62 |
| amazon-photo | 7650 | 119081 | 8 | 0.83 | 0.85 | 0.77 | 0.79 | 0.67 | 0.72 |
| lastfm-asia | 7624 | 27806 | 18 | 0.87 | 0.83 | 0.77 | 0.86 | 0.74 | 0.68 |
| facebook | 22470 | 170823 | 4 | 0.89 | 0.88 | 0.82 | 0.82 | 0.62 | 0.74 |
| github | 37700 | 289003 | 2 | 0.85 | 0.80 | 0.38 | 0.38 | 0.13 | 0.15 |
| twitter-hate | 2700 | 11934 | 2 | 0.78 | 0.67 | 0.50 | 0.55 | 0.23 | 0.51 |
| ogbn-arxiv | 169343 | 1157799 | 40 | 0.65 | 0.64 | 0.42 | 0.59 | 0.45 | 0.53 |
| ogbn-products | 2449029 | 61859012 | 47 | 0.81 | 0.83 | 0.46 | 0.79 | 0.68 | 0.72 |
| actor | 7600 | 26659 | 5 | 0.22 | 0.22 | 0.01 | 0.00 | 0.00 | 0.00 |
| flickr | 89250 | 449878 | 7 | 0.32 | 0.32 | 0.07 | 0.09 | 0.01 | 0.01 |
| deezer-europe | 28281 | 92752 | 2 | 0.53 | 0.53 | 0.03 | 0.03 | 0.00 | 0.00 |
| twitch-de | 9498 | 153138 | 2 | 0.63 | 0.60 | 0.14 | 0.14 | 0.02 | 0.03 |
| twitch-pt | 1912 | 31299 | 2 | 0.57 | 0.59 | 0.12 | 0.11 | 0.01 | 0.02 |
| twitch-gamers | 168114 | 6797557 | 2 | 0.55 | 0.56 | 0.09 | 0.09 | 0.01 | 0.02 |
| genius | 421961 | 922868 | 2 | 0.59 | 0.51 | 0.02 | -0.05 | 0.00 | 0.17 |
| arxiv-year | 169343 | 1157799 | 5 | 0.22 | 0.29 | 0.07 | 0.01 | 0.04 | 0.12 |
| snap-patents | 2923922 | 13972547 | 5 | 0.22 | 0.21 | 0.04 | 0.00 | 0.02 | 0.00 |
| wiki | 1770981 | 242605360 | 5 | 0.38 | 0.28 | 0.17 | 0.15 | 0.06 | 0.04 |
| roman-empire | 22662 | 32927 | 18 | 0.05 | 0.05 | 0.02 | -0.05 | 0.11 | 0.11 |
| amazon-ratings | 24492 | 93050 | 5 | 0.38 | 0.38 | 0.13 | 0.14 | 0.04 | 0.04 |
| minesweeper | 10000 | 39402 | 2 | 0.68 | 0.68 | 0.01 | 0.01 | 0.00 | 0.00 |
| workers | 11758 | 519000 | 2 | 0.59 | 0.63 | 0.18 | 0.09 | 0.01 | 0.02 |
| questions | 48921 | 153540 | 2 | 0.84 | 0.90 | 0.08 | 0.02 | 0.00 | 0.01 |

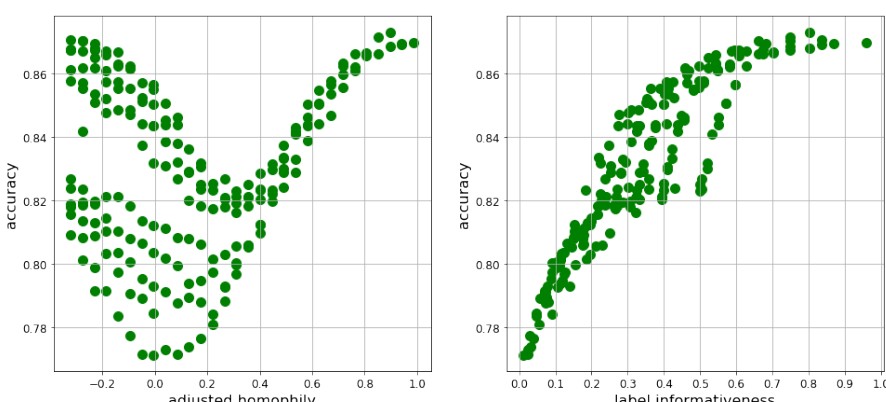

Figure 3: Dependence of GraphSAGE accuracy on homophily and label informativeness for synthetic SBM graphs

edge label informativeness, averaging is over the edges, which implies that high-degree nodes make a larger contribution to the final measure. It is natural to expect that for high-degree nodes the amount of information from each individual neighbor is lower than for low-degree nodes since neighbors of high-degree nodes are expected to be more diverse and closer to the "average" distribution. Thus, it is natural to expect edge label informativeness to often be smaller than node label informativeness, which agrees with Table 5. Note that for most of the real datasets, the values of these two measures are rather close, with `twitter-hate`, `genius`, and `arxiv-year` being the exceptions.

# F Correlation of LI with GNN performance

## F.1 Experimental setup

For our experiments on synthetic data, we select 208 combinations of homophily and LI. For each combination, we generate 10 random graphs with the corresponding homophily and LI values using the SBM-based model described in Section 4.2. Each graph has 1000 nodes (250 nodes for each class) and the expected node degree of 10. Node features are taken from the four largest classes in the `cora` dataset (each of these four classes is mapped to one class in the synthetic data, and node features are sampled randomly from the corresponding class). For each synthetic graph, we create 10 random 50%/25%/25% train/validation/test splits. Thus, for each model, we make 10 runs per graph or 100 runs per homophily/LI combination, totaling 20800 runs.

We use **GCN** [16], **GraphSAGE** [10], **GAT** [43] and **Graph Transformer (GT)** [40] as representative GNN architectures for our experiments. For GraphSAGE, we use the version with the mean aggregation function and do not use the node sampling technique used in the original paper. For all models, we add a two-layer MLP after every graph neighborhood aggregation layer and further augment all models with skip connections [11], layer normalization [2], and GELU activation functions [12]. For all models, we use two graph neighborhood aggregation layers and hidden dimension of 512. We use Adam [15] optimizer with a learning rate of $3 \cdot 10^{-5}$ and train for 1000 steps, selecting the best step based on the validation set performance. We use a dropout probability of 0.2 during training. Our models are implemented using PyTorch [31] and DGL [44].

## F.2 Synthetic data based on SBM model

In Figure 2a of the main text, we show the results for GraphSAGE. In Figure 3, we additionally plot the dependence of GraphSAGE accuracy on homophily and label informativeness. It can be seen that the model's accuracy is much more correlated with label informativeness than with homophily. The results for GCN, GAT, and GT are presented in Figures 4, 5, and 6, respectively. As can be seen, the performance of all models generally follows the same pattern, and LI is a better predictor of model performance than homophily. In particular, when LI is high, all models achieve high accuracy even if homophily is negative. Recall that we provide Spearman correlation coefficients between model accuracy and adjusted homophily or LI in Table 3.

## F.3 Semi-synthetic data from [24]

We use the same experimental setting described above for training GCN, GraphSAGE, GAT, and GT on modifications of the `cora` and `citeseer` graphs from [24]. The results for GraphSAGE and `cora` are provided in Figure 2b of the main text. The results for GraphSAGE and `citeseer` are shown in Figure 7.

## F.4 Synthetic data based on LFR model

We also experiment with the LFR random graph model, which is a well-known benchmark for community detection (we treat communities as node labels). We generated 39 LFR graphs with the mixing parameter values evenly spaced between 0.025 and 0.975. Each graph has 1000 nodes and 5 communities. The maximum community size is 350, and the minimum community size is 100. The average node degree is 10, and the maximum node degree is 50. Note that homophily monotonically changes with the mixing parameter, while LI is $U$-shaped. Similarly to our experiments with SBM, we measure the Spearman correlation coefficient between accuracy and LI / $h_{adj}$, see Table 6. We see that LI is a better predictor of GNN performance than homophily.

Table 6: Spearman correlation between model accuracy and characteristics of LFR graphs

| Model | $h_{adj}$ | $\text{LI}_{edge}$ |
|---|---|---|
| GCN | 0.79 | **0.96** |
| GraphSAGE | 0.68 | **0.99** |
| GAT | 0.77 | **0.97** |
| GT | 0.77 | **0.97** |

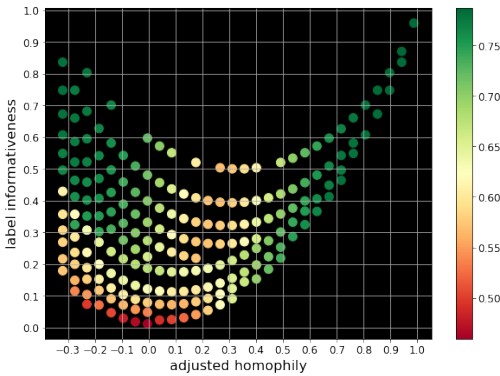

Figure 4: Accuracy of GCN on synthetic SBM graphs

Figure 5: Accuracy of GAT on synthetic SBM graphs

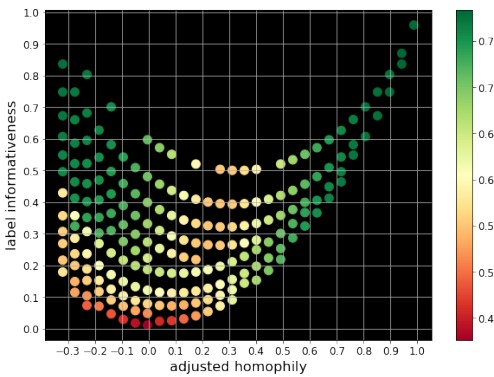

Figure 6: Accuracy of GT on synthetic SBM graphs

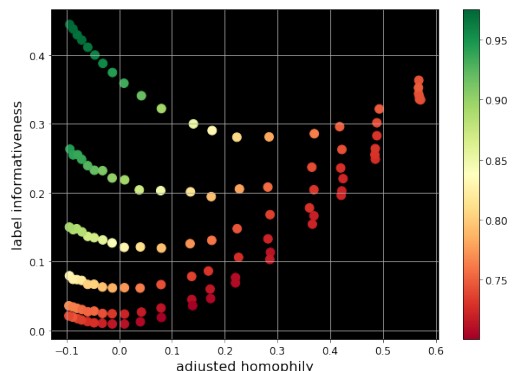

Figure 7: Accuracy of GraphSAGE on semi-synthetic `citeseer` graphs from [24]

### F.5 Synthetic data from [22]

Luan et al. [22] have shown that GNNs can achieve strong performance on certain heterophilous graphs. Again, this phenomenon can be explained by the high label informativeness of the heterophilous graphs used for these experiments.

The authors investigate how different levels of homophily affect GNN performance. They find that the curve showing the dependence of GNN performance on edge homophily (as well as on node homophily) is $U$-shaped: GNNs show strong results not only when edge homophily is high, but also when edge homophily is very low. Our label informativeness explains this behavior. We use the same data generating process as in [22] and find that the curve of label informativeness depending on edge homophily is also $U$-shaped (see Figure 8). Thus, on the datasets from [22], GNNs perform well exactly when label informativeness is high regardless of edge homophily. The $U$-shape of the label informativeness curve is not surprising: when edge homophily is very low, knowing that a node has a neighbor of a certain class provides us with information that this node probably *does not belong* to this class.

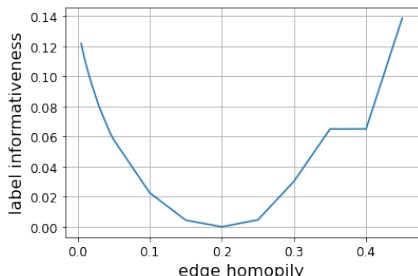

Figure 8: Label informativeness depending on edge homophily on synthetic graphs from [22]

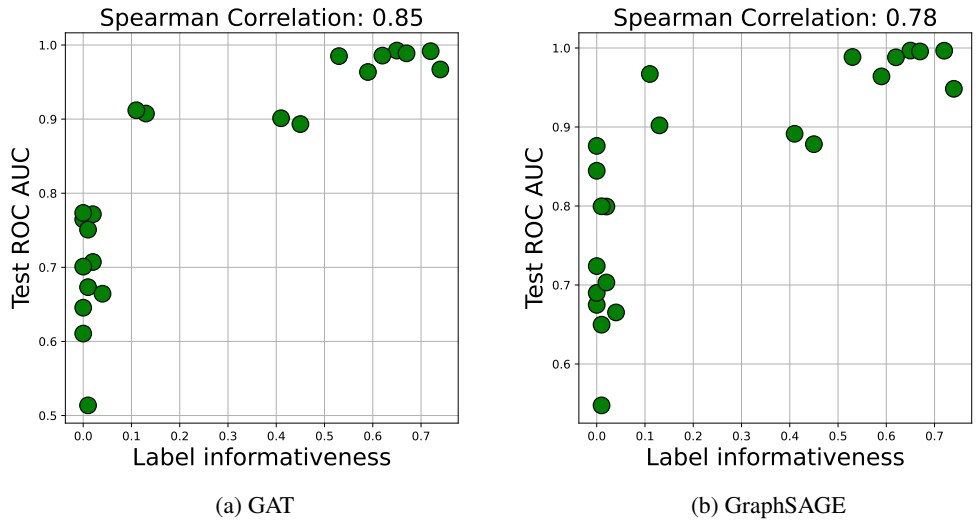

Figure 9: Performance of GAT and GraphSAGE on real graphs vs $\text{LI}_{edge}$

Table 7: Spearman correlation coefficients between ROC AUC score and characteristics of real-world datasets from Table 5

| Model | $h_{edge}$ | $h_{node}$ | $h_{class}$ | $h_{adj}$ | $\text{LI}_{edge}$ | $\text{LI}_{node}$ |
|---|---|---|---|---|---|---|
| GraphSAGE | 0.64 | 0.62 | 0.67 | 0.61 | 0.78 | **0.82** |
| GAT | 0.69 | 0.65 | 0.74 | 0.69 | **0.85** | 0.84 |

### F.6    Correlation of GNN performance with homophily and LI on real datasets

Analyzing which measures better agree with GNN performance on real data is non-trivial. Indeed, if we consider diverse node classification problems, there are many factors affecting the GNN performance: the size of the datasets and fraction of training examples, the edge density, the number of features and their informativeness, the number of classes and their balance, and so on. However, as requested by the reviewers, we analyze how well different homophily measures and LI correlate with GNN performance on the datasets in Table 5. In this experiment, we measure the ROC AUC score since some of the datasets are highly unbalanced, and thus accuracy is not a suitable performance measure. For the datasets with more than two classes, we report the macro-averaged ROC AUC score. The relation between ROC AUC and LI for GAT and GraphSAGE is shown in Figure 9. The Spearman correlation for all the measures is reported in Table 7. We see that the largest correlation is achieved by $\text{LI}_{edge}$ and $\text{LI}_{node}$. Let us note that this table does not aim to compare homophily measures with each other since a good homophily measure is expected to properly capture the tendency of edges to connect similar nodes. Thus, we compare homophily measures using the proposed theoretical framework in Section 2.

