# OpenReview forum: "Characterizing Graph Datasets for Node Classification: Homophily-Heterophily Dichotomy and Beyond"
_NeurIPS.cc/2023/Conference — NeurIPS 2023 poster_

### Official Review · Reviewer_oQPA · 2023-06-26

**Soundness:** 3 good
**Presentation:** 3 good
**Contribution:** 2 fair
**Rating:** 3
**Confidence:** 4

**Summary:**

The main contribution of this paper is a new measure called label informativeness that characterizes how much information a neighbor’s label provides about a node’s label. The paper also discusses the expected properties of homophily. The experimental results are based on the relation between the performance of GNN and the proposed metric.

**Strengths:**

1. The quality of the theoretical demonstrations. Most of the demonstrations seem correct, even though I did not go exhaustively through the details, as they are in the supplementary material.

2. Good discussion about homophily. The paper covers several aspects of homophily, mentioning the desired properties of this measure.

3. The clarity of the proposition. It is clearly stated the main contribution of the paper.

4. Readability. The paper is well-written, being very simple to read.


**Weaknesses:**

1. The novelty of the work could be questionable. The measure is not novel, please refer to "On the Estimation of Relationships Involving Qualitative Variables", available at https://www.jstor.org/stable/2775440. The submitted paper presents an uncertainty coefficient that is closely related to the work proposed in this paper. Even though, that paper does not focus on graphs, the proposed measure seems to be the application of this measure over a particular distribution, rather than the proposition of a new measure. Second, the same formulas and equations are already published in [16]. Moreover, in [16], it is mentioned that LI was previously introduced.

2. The conclusion mentions "LI characterizes how much information a neighbor’s label provides about a node’s label". Unfortunately, this is not explained in the paper, except by the phrase "$LI \in [0, 1]$ If the label y_n allows for unique reconstruction of y_\epsilon , then LI = 1. If y_\epsilon and y_n are independent, LI = 0".

3. The paper mentions that "Through a series of experiments, we show that LI correlates well with the performance of GNNs.". Based on https://www.jstor.org/stable/2775440, I believe that this is related because the measure is based on the classification performance, rather than GNN.

4. The limitations mention "this measure to be both informative and simple to compute and interpret.". However, no interpretations are given besides the values of 0 and 1.

**Questions:**

1. Regarding the analysis of the value. Does LI have a linear scale of the values? Does a value of 0.5 mean 50% of reconstruction?

2. Is the LI general enough to obtain similar correlations with other classification models (I think so)?

3. Could you explain the main difference between LI (the metric, not the way that is used) and https://www.jstor.org/stable/2775440?

4. Could you explain the difference between LI with respect to the LI metric used in the published paper [16]?

**Limitations:**

Yes, the authors clearly state the limitations of their work. However, there are some limitations that were omitted.

As a suggestion, if the main contribution is the use of the JSTOR paper in graphs. Please state it this way, rather than saying that this is a new measure.

---

> ### Author Rebuttal · Authors · 2023-08-09
>
> Thank you for the review and comments! In this response, we address your questions and concerns.
>
> **Contribution of our work**
>
> In our discussion below we argue why we think that LI is a novel graph characteristic and how it relates to known concepts in information theory. However, in this discussion, we do not want to narrow the scope of our paper to only one formula (3), thus let us briefly state the main contributions of our work. We think that our main contributions are the proposed theoretical framework and theoretical analysis of different measures within this framework. We believe that the important takeaway from our work is that one has to be careful when choosing graph characteristics to describe and compare graph datasets. In particular, it is important to verify the unbiasedness of the measures. We advise using adjusted homophily and LI since both these measures are unbiased and have other important desirable properties.
>
> **The novelty of LI and its relation to concepts from information theory**
>
> In our work, we use normalized mutual information (equation (3)) as a well-known and widely used concept and therefore do not cite any works here. Thanks for pointing out that we do not explicitly call this measure “normalized mutual information” (or “coefficients of constraint”, “uncertainty coefficient”, or “proficiency” which are other terms used in the literature) -- this happened because we tried to explain the intuition and interpretation of this measure while introducing it step-by-step. This will be fixed - we will explicitly write that the concept is well-known.
> However, the main novelty of LI is applying this known notion of mutual information to graph datasets. For this, we need to define graph-based random variables to which this concept can be applied. Note that such random variables are not predetermined and there are several options, some of which we discuss in the paper. We are not aware of prior work applying the concept of mutual information to graphs in a similar manner to characterize edge-label relations.
> We address other questions and concerns below.
>
> > Could you explain the main difference between LI (the metric, not the way that is used) and https://www.jstor.org/stable/2775440?
>
> As described above, we use the uncertainty coefficient as a known concept in information theory (and will explicitly state it in the paper to avoid any possible misunderstanding). However, to properly address the questions on the relation to Theil (1970), we would like to kindly ask if you can point us to a particular equation or definition in Theil (1970) to which we should compare our notion of LI.
>
> Also, we cannot agree with the distinction between the metric and the way that it is used, since in our case defining the random variables is a part of the metric definition. Drawing parallels with other known graph characteristics, one could say that the assortativity coefficient is the application of Pearson correlation to graphs, edge homophily is the application of the operation “average” to graphs, PageRank is the application of eigenvectors to graphs, and so on. In all these cases, the characteristics are defined by the application of a known concept to a graph domain.
>
> > The conclusion mentions "LI characterizes how much information a neighbor’s label provides about a node’s label". Unfortunately, this is not explained in the paper
>
> The cited sentence refers to the concept of information from the information theory, i.e., this is just a rephrasing of the definition of LI: by definition, LI describes the (relative) amount of information revealed by the label of a single neighbor. We provide more intuition in our answer to the next question.
>
> > Regarding the analysis of the value. Does LI have a linear scale of the values? Does a value of 0.5 mean 50% of reconstruction?
>
> The value of LI can be related to the cross-entropy loss. Indeed, assume that no information about the neighbors’ labels is available. Then, the best classifier in terms of the cross-entropy assigns probability $p(k)$ to each class $k$. For this classifier, the cross-entropy loss is the entropy of $p(\cdot)$. If we know the label of one neighbor, the cross-entropy loss of the best possible classifier is reduced to the conditional entropy. Thus, we can say that LI describes the relative reduction of the expected cross-entropy loss of the best possible classifier. So, the value of 0.5 means that this loss becomes two times smaller. We hope that this explanation addresses the concern about the interpretability of the measure.
>
> This simple intuition, however, does not account for the fact that in graphs each node has several neighbors which may have various distribution of labels. Thus, the reasoning above cannot be directly applied, so we additionally verify the usefulness of LI as a graph characteristic via experiments.
>
> > Is the LI general enough to obtain similar correlations with other classification models (I think so)?
>
> We hope that our reply to the previous question addresses this concern. But if no, could you please clarify which classification models we need to consider?
>
> > Could you explain the difference between LI with respect to the LI metric used in the published paper [16]?
>
> Since explaining the current paper's relationship to LI used in [16] involves a potential double-blind issue, we have checked it with the AC, and the AC confirms that the LI in [16] does not negate the novelty of the LI in the current work.
>
> We hope that our response addresses your concerns. We are happy to answer any additional questions during the discussion period if needed.

---

> > ### Comment · Reviewer_oQPA · 2023-08-17
> > **Thanks for the answer**
> >
> > I have read the author response. Thanks for agreeing that the measure is similar to previous work. This was one of my main issues.

---

> > > ### Author Response · Authors · 2023-08-18
> > >
> > > Thank you for reading our response! This concern will be addressed in the text by explicitly saying that we use normalized mutual information as an ingredient of our definition. We hope that our response to other questions clarifies the intuition behind LI and its scale. We also thank the reviewer for acknowledging the quality of the theoretical analysis, discussion about homophily, and readability of the paper in the review.

---

### Official Review · Reviewer_kh3x · 2023-07-02

**Soundness:** 3 good
**Presentation:** 4 excellent
**Contribution:** 3 good
**Rating:** 6
**Confidence:** 3

**Summary:**

This paper analyzes homophily measures and proposes a new characteristic highly correlated with performance. Specifically, the authors suggest desirable properties for the metric and show that current homophily measures do not satisfy some important properties, so suggest a new homophily measure to satisfy them. Furthermore, beyond homophily arguments, the authors suggest a new characteristic that is more highly correlated with performance from the perspective of the usefulness of neighbor labels to predict the label of the corresponding center node. They demonstrate that the correlation between the values of proposed measures and performance is significantly higher than current homophily measures on various architectures and datasets.

**Strengths:**

- The authors suggest the desirable properties for the metrics and show that current homophily measures do not satisfy important traits such as asymptotic constant baseline.
- They suggest adjusted homophily to satisfy most of the important properties including the asymptotic constant baseline.
- They propose a new characteristic describing the effectiveness of graph structure for performance.
- The proposed characteristic exhibits a significantly higher correlation with performance compared to homophily measures.

**Weaknesses:**

Although the authors provide the correlations between performance and a given metric on Cora and Citeseer by modifying graphs synthetically, it seems not enough to show the general applicability.

**Questions:**

Could you provide the results when calculating the correlation only based on the values of metrics and performance in Appendix Table 4, without manipulating the original graphs?

**Limitations:**

As the authors mentioned, LI only considers node label interaction in graphs.

---

> ### Author Rebuttal · Authors · 2023-08-09
>
> Thank you for the review and positive feedback! In our experiments we indeed mostly focus on synthetic datasets where connectivity patterns can be easily controlled and on semi-synthetic data previously used in the literature for the analysis of homophily measures. Originally, we have not experimented with real datasets since various factors besides edge-label relations may affect the performance. Moreover, we did not want to move the focus of our work from theoretical analysis to only experimental results - LI is a very simple measure that cannot pretend to capture all complex situations that can arise in practice.  However, as asked by several reviewers, we also measured the correlation between performance and different measures on real datasets from Table 4, please see our [general response](https://openreview.net/forum?id=m7PIJWOdlY&noteId=hxPviGCfUx). Here we show that even on real datasets LI better agrees with GNN performance than homophily measures. We hope these additional experiments address your concern and will be happy to answer any additional questions!

---

> > ### Comment · Reviewer_kh3x · 2023-08-12
> > **Response to Authors**
> >
> > Thank you for your response. I read your response and PDF carefully. I'll keep my score as weak accept.

---

### Official Review · Reviewer_eRJB · 2023-07-05

**Soundness:** 3 good
**Presentation:** 3 good
**Contribution:** 3 good
**Rating:** 7
**Confidence:** 4

**Summary:**

This paper focuses on developing techniques to characterize graph node classification datasets. First, this paper has proposed a theoretical framework to develop measures of graph’s characteristics. Then two measures called “adjusted homophily” and “label informativeness” are proposed to easily characterize the connectivity patterns of graph datasets. And experimental results show the stronger correlation of proposed measures with GNNs’ performance.

**Strengths:**

1.	This paper has analysis the drawbacks of existing measures, like node homophily and edge homophily.
2.	This paper has proposed a generic framework with theoretical analysis to develop techniques for characterizing graph node classification datasets, which will help to design more suitable models on certain datasets. And proposed minimal agreement, maximal agreement, and monotonicity is very useful.
3.	The proposed adjusted homophily get rid of the number of class and class size on different datasets and is of broader applicability.
4.	The proposed label informativeness have a strong correlation with the GNN’s performance, which is interesting and novel. Moreover, this can help to design more powerful development of heterophily-suited GNNs.


**Weaknesses:**

1.	Only Cora and Citeseer that be considered as homophilic graphs are used for showing the correlation between GNNs’ performance and LI, more heterophilic graphs should be tested.
2.	Assortativity coefficient is mentioned in section 2.4, but there is not comparison with proposed measures.


**Questions:**

1.	More details about how to get the adjusted value in lines 183 and 184.
2.	Edge label informativeness seems to be smaller than node label informativeness in most cases, any explanation about this?


**Limitations:**

Authors have adequately addressed the limitations in this paper.

---

> ### Author Rebuttal · Authors · 2023-08-09
>
> Thank you for the review and for acknowledging the usefulness of our theoretical framework! We address your questions and concerns below.
>
> > Only Cora and Citeseer that be considered as homophilic graphs are used for showing the correlation between GNNs’ performance and LI, more heterophilic graphs should be tested.
>
> In our experiments we mostly focus on synthetic datasets where connectivity patterns can be easily controlled. Here we consider both SBM and LFR random graph models with varying parameter combinations allowing us to get non-trivial patterns. We also use semi-synthetic data from [17], where datasets are generated based on Cora and Citeseer. Originally, we have not experimented with real datasets since various factors besides edge-label relation may affect the performance. However, as asked by several reviewers, we also measured the correlation between performance and different measures on real datasets from Table 4, please see our [general response](https://openreview.net/forum?id=m7PIJWOdlY&noteId=hxPviGCfUx). Here we show that even on real datasets LI better agrees with GNN performance than homophily measures. We hope these additional experiments address your concern.
>
> > Assortativity coefficient is mentioned in section 2.4, but there is not comparison with proposed measures.
>
> As written in Section 2.4, the assortativity coefficient reduces to (2) when applied to discrete node attributes on undirected graphs. Thus, it produces the same values in our setting. We call this measure _adjusted homophily_ in our paper for the following reasons:
> - To avoid ambiguity with other usages of assortativity, e.g., assortativity coefficient often refers to degree-degree correlations in graphs without node labels
> - To emphasize that adjusted homophily is a standard adjustment for chance applied to edge homophily (i.e., adjusted homophily relates to edge homophily as, e.g., adjusted Rand index relates to Rand index)
> - To emphasize that this measure indicates the level of homophily since this coefficient is rarely used in graph ML literature for this purpose
>
> > More details about how to get the adjusted value in lines 183 and 184.
>
> According to the configuration model, the probability that a given edge goes to a node of degree $d$ equals $\frac{d}{2|E|}$, where the denominator equals the sum of all node degrees. Then, to get the probability of connection to a node from class $k$, we need to sum these values over all nodes from class $k$. Thus, we obtain $\frac{D_k}{2|E|}$ since $D_k$ is the sum of all degrees for nodes from class $k$. Multiplying this by $D_k$, we get twice the expected number of edges going from class $k$ to class $k$. Dividing this by $2|E|$ gives that $\frac{D_k^2}{4|E|^2}$ is the expected fraction of edges going from class $k$ to class $k$. Then, if we sum over all the classes, we get the expected number of homophilous edges. We subtract this value from $h_{edge}$ to get an unbiased measure. There are some negligible error terms in this reasoning, so we refer to section B.4 where all error terms are carefully tracked.
>
> > Edge label informativeness seems to be smaller than node label informativeness in most cases, any explanation about this?
>
> Edge and node label informativeness differ in how they weight high/low-degree nodes. For node label informativeness, averaging is over the nodes, so all nodes are weighted equally. For edge label informativeness, averaging is over the edges, which implies that high-degree nodes make a larger contribution to the final measure. It is natural to expect that for high-degree nodes the amount of information from each individual neighbor is lower than for low-degree nodes since neighbors of high-degree nodes are expected to be more diverse and closer to the “average” distribution. Thus, edge LI is expected to be smaller. Thank you very much for this observation, we will add this discussion to the text!
>
> We hope that we addressed your concerns and will be happy to answer any additional questions!

---

> > ### Comment · Reviewer_eRJB · 2023-08-16
> >
> > Thank you for the response. I have read all other reviews and rebuttal, and submitted rebuttals have addressed most concerns. I have raised my score.

---

### Official Review · Reviewer_QJti · 2023-07-06

**Soundness:** 3 good
**Presentation:** 4 excellent
**Contribution:** 2 fair
**Rating:** 6
**Confidence:** 3

**Summary:**

The paper proposes new measures to characterize homophily and heterophily in graph datasets. First, some desirable properties are asserted for a good measure of homophily. It is discussed whether various prior measures of homophily satisfy these properties, then a new measure called *adjusted homophily* is proposed; this is a modification of the established measure of *edge homophily* using information about (degree-adjusted) class distributions of nodes. After this, to capture the fact that a node's class label can be, depending on the dataset, either informative or uninformative about its neighbor's class label, a measure called *label informativeness* (LI) is also proposed. When randomly sampling an edge $(u,v)$, this is the fraction of the information content of a $u$'s class label that is given by knowing $v$'s class label. After establishing these measures, the paper lists them for several real-world datasets. In experiments on synthetic SBM graphs and semi-synthetic modifications of the Cora graph, the authors show that LI is better correlated with GNN performance than homophily.

**Strengths:**

- The homophily/heterophily topic has seen a lot of interest in the past few years, so this work is likely to get much attention.
- The paper is very well-written. It is logically organized and easy to read.
- The paper makes a compelling case that both the adjusted homophily and LI measures are the most sensible such measures.

**Weaknesses:**

- While LI seems like a very sensible measure, it is rather straightforward, so I am left thinking: First, has it really not been proposed by some prior work, especially given that the general concept of normalized information measures is well-established (e.g., [this](https://en.wikipedia.org/wiki/Mutual_information#Normalized_variants))? I don't see this as a weakness on it's own, since I am not aware of such prior work. However, I do feel that some more graph-specific theoretical work addressing the first limitation listed in the paper (i.e., variants of homophily/LI that consider multiple hops in the graph) would significantly increase the contribution. The paper seems to state that this is beyond the scope of the work, but I feel the amount of contribution is a bit borderline as it is.
- The experiments positing that LI is well correlated with GNN performance are on synthetic or semi-synthetic datasets only. Ideally, some trend could be shown across a variety of real-world datasets, though it is understandable that such a trend may not exist given various factors besides LI.


### Typos
- Figure 8: "edge homopily"

**Questions:**

My questions correspond to points from the prior section:
- Assuming that LI is indeed a new measure, do you see some variant of it that could work for more than 1 hop across the graph? Or otherwise, do you think such measures are unlikely to be informative?
- Do you expect there to be a general trend (even if it is a weak trend) that, across the usual benchmark graph datasets, higher LI corresponds to higher node classification performance? Or do you at least expect a higher correlation than with (adjusted) homophily? I think it would be interesting to see plots like the ones for the synthetic datasets, except for (some fraction of) the graphs in Table 4.

**Limitations:**

Limitations are adequately addressed in a designated section of the conclusion.

---

> ### Author Rebuttal · Authors · 2023-08-09
>
> Thank you very much for your positive feedback and useful comments! We address your concerns below.
>
> > While LI seems like a very sensible measure, it is rather straightforward, so I am left thinking: First, has it really not been proposed by some prior work, especially given that the general concept of normalized information measures is well-established
>
> We indeed use the normalized mutual information as a well-known concept when defining the LI measure. Our suggestion here is to apply this concept to graphs by defining random variables to which it should be applied. We are not aware of prior work using normalized mutual information for graphs in a similar manner to characterize edge-label relations.
>
> > Assuming that LI is indeed a new measure, do you see some variant of it that could work for more than 1 hop across the graph? Or otherwise, do you think such measures are unlikely to be informative?
>
> This is a very reasonable suggestion. At earlier stages of our research, we indeed defined and measured “two-hop homophily” and “two-hop LI”. These are similar measures to those discussed in the paper but defined via pairs of nodes at distance two (or larger distances can be considered). However, we have not observed any interesting patterns for these measures: non-trivial connection patterns become harder to detect as we increase distances between nodes. Thus, we decided to make our work more focused and to advise simple but efficient one-hop measures.
>
> > Do you expect there to be a general trend (even if it is a weak trend) that, across the usual benchmark graph datasets, higher LI corresponds to higher node classification performance? Or do you at least expect a higher correlation than with (adjusted) homophily? I think it would be interesting to see plots like the ones for the synthetic datasets, except for (some fraction of) the graphs in Table 4.
>
> You are absolutely right that various factors besides LI may affect the performance - and it is the reason why we initially experimented only on synthetic and semi-synthetic datasets. However, to address this question, we also conducted the experiment you suggest and measured the correlation on the datasets in Table 4. We report the results in our [general response](https://openreview.net/forum?id=m7PIJWOdlY&noteId=hxPviGCfUx). Here we show that despite other factors potentially affecting the results, LI better agrees with GNN performance than homophily measures.
>
> We hope that we addressed your concerns and will be happy to answer any additional questions!

---

> > ### Comment · Reviewer_QJti · 2023-08-15
> >
> > Thank you for the response. While I am still a bit doubtful on the overall amount of contribution / novelty of the introduced measure, I think the new experimental result is a good argument for LI, and I have raised my score.
> >
> > Edit: I see that Reviewer oQPA has the same doubts as me regarding novelty. While I won't lower my score, I do think that the level of contribution here may be more suitable for a workshop paper rather than the main conference; the contribution would be improved if the authors added content by finding and analyzing some interesting pattern with a more graph-theoretic measure, e.g., a multi-hop one as we discussed above.

---

### Author Rebuttal · Authors · 2023-08-09

We would like to thank all the reviewers for their comments and suggestions! In this general response, we address the comment raised by several reviewers about conducting additional experiments on real datasets.

Analyzing which measures better agree with GNN performance on real data is non-trivial. Indeed, if we consider diverse node classification problems, there are many factors affecting the GNN performance: the size of the datasets and fraction of training examples, the edge density, the number of features and their informativeness, the number of classes and their balance, and so on.

However, for the completeness of the study, we analyze how well different homophily measures and LI correlate with GNN performance on real datasets from Table 4. In this experiment, we measure the ROC AUC score since some of the datasets are highly unbalanced, and thus accuracy is not a suitable performance measure. For the datasets with more than two classes, we report the macro-averaged ROC AUC score.

The relation between ROC AUC and LI (for GAT and GraphSAGE) is shown in Figures 1 and 2 in the attached pdf. Table 1 reports the Spearman correlation of ROC AUC with all the measures. We see that the largest correlation is achieved by LI (node and edge variants).

Let us note that this experiment does not aim to compare homophily measures with each other since a good homophily measure is expected to properly capture the tendency of edges to connect similar nodes and is not expected to correlate with GNN performance. That is why we compare homophily measures using the proposed theoretical framework in our paper.

We hope that these experiments address the concerns of the reviewers and we are open to further discussions.

---

### Decision · Program_Chairs · 2023-09-21

**Decision:**

Accept (poster)

**Comment:**

The paper makes a number of novel contributions:
* Discuss desirable properties for homophily measures, and show that existing homophily measures fail to satisfy some of these
* Studies the "adjusted homophily" measure and show the properties it satisfies
* Propose the label informativeness (LI) measure, which aims to go beyond the homophily-heterophily dichotomy by measuring how informative a node's labels are about its neighbor's labels; show that it satisfies some of the properties
* Empirically show the correlation between LI and GNN performance

Overall, most reviewers are on the positive side, with 1 on the negative side mainly due to concerns with novelty. Specifically, there is the concern due to similarity to "On the Estimation of Relationships Involving Qualitative Variables" (https://www.jstor.org/stable/2775440). This paper is about information theoretic measures in a general (non-graph) context. Use of information theoretic measures is indeed classical and standard, and the contributions of the current work should be considered in light of that, but my view is the specific paper https://www.jstor.org/stable/2775440 does not significantly affect the consideration beyond this. In my view, it is a good thing that LI is designed in a simple way based on standard information theoretic measures, since it is better for metrics to have a clear and easily understood interpretation, rather than being complex, and the novelty is more in the points listed above rather than the specific design choices of the LI metric.

The paper [16] is also noted by the reviewer, but discussion (between AC / SAC) confirmed that the LI in [16] does not negate the novelty of the LI in the current work.

Moreover, the author response adds results showing that LI correlates better with GNN performance on real datasets (rather than just synthetic datasets) compared to existing homophily measures, which is interesting. Overall, the paper provides detailed empirical and theoretical analysis on the important topic of heterophily, which provides some insights which are not well known in the GNN literature. Hence, I recommend acceptance.

Authors are requested to note improvements / action items arising from discussion with the reviewers, such as adding the new empirical results to the manuscript, clarifications to the paper arising from reviewer discussion (e.g. line 183-184), etc.